# Factored Causal Representation Learning for Robust Reward Modeling in RLHF

**Yupei Yang** [1 2 †] **Lin Yang** [2] **Wanxi Deng** [2] **Lin Qu** [2] **Fan Feng** [3 4] **Biwei Huang** [3] **Shikui Tu** [1 *] **Lei Xu** [1]

## Abstract

A reliable reward model is essential for aligning large language models (LLMs) with human preferences through reinforcement learning from human feedback (RLHF). However, standard reward models are susceptible to spurious features that are not causally related to human labels. This can lead to *reward hacking*, where high predicted reward does not translate into better behavior. In this work, we address this problem from a causal perspective by proposing a factored representation learning framework that decomposes the model's contextual embedding into (1) causal factors that are sufficient for reward prediction and (2) non–causal factors that capture reward-irrelevant attributes such as length or sycophantic bias. The reward head is then constrained to depend only on the causal component. In addition, we introduce an adversarial head trained to predict reward from the non-causal factors, while applying gradient reversal to discourage them from encoding reward-relevant information. Experiments on both mathematical and dialogue tasks demonstrate that our method learns more robust reward models and consistently improves downstream RLHF performance over state-of-the-art baselines. Analyses on length and sycophantic bias further validate the effectiveness of our method in mitigating reward hacking behaviors.

## 1. Introduction

In recent years, RLHF has emerged as a powerful approach for aligning LLMs with human preferences (Ouyang et al., 2022; Bai et al., 2022). A core component of RLHF is the reward model, which serves as a proxy for human judgment during policy optimization. However, standard reward model training is prone to learning spurious correlations: the model may assign higher scores to preference-irrelevant patterns, such as response length or sycophantic phrasing, rather than the true causal drivers of human preference (Liu et al., 2024; Miao et al., 2024; Wang et al., 2025). The LLM then exploits these shortcuts during RLHF, achieving higher predicted rewards while drifting from the intended objective—a phenomenon known as reward hacking (Amodei et al., 2016; Gao et al., 2023b; Skalse et al., 2022).

To tackle this issue, extensive efforts have been made to mitigate the influence of known spurious factors in reward modeling. For example, ODIN (Chen et al., 2024) decomposes the reward head into separate quality and length components to reduce reward hacking driven by response length. Park et al. (2024) develops a length-regularization strategy to prevent length exploitation during DPO (Rafailov et al., 2023). Wang et al. (2025) further proposes a more general maximum mean discrepancy (MMD)-based regularization that can constrain spurious behaviors beyond length, such as sycophantic bias (Sharma et al., 2023) or concept bias (Zhou et al., 2024). Despite their effectiveness, these approaches require explicitly specifying the spurious variables to be controlled, whereas it is challenging to anticipate all possible exploitation patterns in practice.

Another line of work seeks to improve reward models by filtering out irrelevant information directly through representation learning. Nath et al. (2024) employs a contrastive objective to learn goal-conditioned representations within the reward model, which helps distinguish between preferred and dispreferred responses. InfoRM (Miao et al., 2024; 2025a) adopts an information-theoretic perspective, introducing a variational information bottleneck objective to encourage the latent representations to retain only information relevant to human preference. However, to the best of our knowledge, none of the existing methods explicitly disentangle reward-irrelevant factors from the latent state for reward modeling. Nevertheless, causal representation learning (Schölkopf et al., 2021; Huang et al., 2022; Kong et al., 2023; Yang et al., 2024b) has demonstrated strong promise in traditional reinforcement learning for learning minimal sufficient state representations that capture only the causally relevant aspects of the environment.

---

[†]This work was done when the author was a research intern at Alibaba Group. [1]Shanghai Jiao Tong University [2]Alibaba Group [3]University of California San Diego [4]Mohamed bin Zayed University of Artificial Intelligence. Correspondence to: Shikui Tu <tushikui@sjtu.edu.cn>.

*Proceedings of the $43^{rd}$ International Conference on Machine Learning*, Seoul, South Korea. PMLR 306, 2026. Copyright 2026 by the author(s).

Motivated by these insights, we propose **CausalRM**, a novel framework that explores the potential of causal representation learning to mitigate spurious correlations in reward modeling. Specifically, CausalRM decomposes the model's contextual embedding into two disentangled components: (1) *causal factors* that are sufficient for reward prediction, and (2) *non-causal factors* that capture reward-irrelevant attributes such as response length or stylistic bias. Building on this factorized representation, we constrain the reward model to predict rewards using only the causal factors, and optimize it with a standard pairwise preference loss augmented with mutual-information-based constraints. To further discourage the non-causal factors from carrying reward-predictive signals, we introduce an adversarial head trained to predict reward from these factors. Following Ganin & Lempitsky (2015), we then optimize it with a gradient reversal layer (GRL), so that the adversarial head learns to predict reward while the encoder is pushed to remove reward-relevant information from the non-causal component.

To summarize, our main contributions are three-fold:

- We investigate the potential of causal representation learning for mitigating reward hacking in RLHF, and propose **CausalRM**, a framework that explicitly decomposes the latent representation into causal factors and reward-irrelevant non-causal factors.

- To characterize both causal and non-causal representations, we design a novel VAE-based architecture with (1) a reward prediction head trained with a pairwise preference loss augmented by mutual information constraints, and (2) an adversarial head trained via gradient reversal to suppress reward-relevant signals in the non–causal component, which jointly enforces sufficiency, minimality, and causal invariance in the learned representations.

- Extensive experiments on mathematical and dialogue tasks show that CausalRM improves both reward model accuracy and downstream RLHF performance. Furthermore, we demonstrate strong mitigation of reward hacking behaviors through reduced sensitivity to length and sycophantic bias, providing empirical validation of causal invariance in learned representations.

## 2. Preliminaries

**Reward modeling in RLHF.**   In RLHF, reward modeling aims to learn a scalar-valued function that approximates human preferences. Given a prompt $x$ and a response $y$, a reward model outputs a scalar reward $r_\theta(x, y)$ indicating the degree of preference for $y$ under $x$.

A widely used formulation is the Bradley–Terry model (Bradley & Terry, 1952), which defines the probability that a preferred response $y^w$ is favored over a rejected response $y^l$ as:

$$p(y^w \succ y^l \mid x) = \frac{\exp(r_\theta(x, y^w))}{\exp(r_\theta(x, y^w)) + \exp(r_\theta(x, y^l))}. \tag{1}$$

Given a human preference dataset $\mathcal{D} = \{(x_i, y_i^w, y_i^l)\}_{i=1}^N$, the reward model is typically trained by minimizing the pairwise negative log-likelihood:

$$\mathcal{L}_{\text{RM}}(\theta) = -\mathbb{E}_{(x, y^w, y^l) \sim \mathcal{D}} \left[ \log \sigma\big(r_\theta(x, y^w) - r_\theta(x, y^l)\big) \right], \tag{2}$$

where $\sigma(\cdot)$ denotes the sigmoid function.

**Standard parameterization and reward hacking.**   In practice, the reward model is often initialized from a supervised fine-tuned (SFT) language model by reusing its backbone as a feature extractor and attaching a lightweight reward head, which is typically a single linear layer. Concretely, given a prompt–response pair $(x, y)$, the SFT backbone produces a contextual representation

$$h = f_\phi(x, y), \tag{3}$$

and the reward head maps it to a scalar reward

$$r_\theta(x, y) = g_\psi(h), \qquad \theta = (\phi, \psi), \tag{4}$$

where $f_\phi$ denotes the pretrained backbone and $g_\psi$ denotes the reward head.

While effective, this parameterization can be vulnerable to reward hacking: the reward model may inherit and amplify preference-irrelevant signals already encoded in the SFT representation, causing the learned reward to correlate with spurious patterns rather than the causal drivers of human judgment (Miao et al., 2024; Wang et al., 2025).

## 3. Methodology

In this section, we first analyze why standard reward models tend to learn spurious correlations from a causal perspective. Building on these insights, we construct a causal reward model, termed CausalRM, that filters out reward-irrelevant information via factored representation learning. Finally, we present the complete estimation procedure for CausalRM.

### 3.1. A Causal View of Reward Hacking

The causal structure underlying standard reward modeling can be represented by Figure 1, where $(x, y)$ are the input prompt and response pair, $z^c$ denotes causal factors that contain the essential information to predict the reward $r$, and $z^{nc}$ represents spurious factors that do not causally influence the true reward, such as response length or stylistic bias.

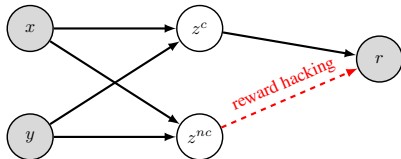

*Figure 1.* Causal graph for standard reward modeling. The prompt–response pair $(x, y)$ encode both causal ($z^c$) and non-causal ($z^{nc}$) factors, which in turn affect the predicted reward $r$. While the path $z^c \rightarrow r$ is desired, the spurious path $z^{nc} \rightarrow r$ leads to reward hacking.

As illustrated, the presence of a direct edge $z^{nc} \rightarrow r$ allows spurious features to directly affect the learned reward, thereby leading to reward hacking. For example, suppose $z^{nc}$ captures response length on mathematical tasks, then changing the length alone may substantially alter the predicted reward, even when the underlying solution quality remains unchanged. Instead, a robust reward model should satisfy *causal invariance* with respect to non-causal factors (Bühlmann, 2020; Veitch et al., 2021):

$$r_\theta(x, y) \perp\!\!\!\perp z^{nc}. \tag{5}$$

In other words, *the reward value should be insensitive to non-causal attributes of the prompt-response pair.*

### 3.2. CausalRM: Factored Causal Representation Learning for Reward Models

Motivated by the causal analysis in Section 3.1, we propose CausalRM, a reward modeling framework that aims to block the spurious path $z^{nc} \rightarrow r$ by (1) structurally restricting reward prediction to depend only on a causal representation, and (2) actively removing reward-predictive information from the remaining representation. Concretely, CausalRM augments a standard RM with a latent-variable bottleneck and two auxiliary heads, jointly promoting *sufficiency* for reward prediction, *invariance* to non-causal variation, and *non-degenerate* representations.

**Factored latent representation.** Given a prompt–response pair $(x, y)$, we first compute a contextual embedding $h = f_\phi(x, y)$ using the SFT backbone. CausalRM then maps $h$ to two latent variables via a variational encoder:

$$q_\alpha(z^c \mid h), \qquad q_\alpha(z^{nc} \mid h), \tag{6}$$

where $z^c$ is encouraged to retain information that is sufficient for predicting human preference, while $z^{nc}$ is encouraged to capture reward-irrelevant attributes. We parameterize both posteriors as diagonal-covariance Gaussians whose means and log-variances are produced by separate linear projections applied to $h$, with standard normal priors $p(z^c) = p(z^{nc}) = \mathcal{N}(0, I)$.

This factorization provides a convenient interface for imposing causal invariance: downstream prediction modules

can be forced to condition only on $z^c$, while $z^{nc}$ serves as a dedicated channel for non-causal variation.

**Causal reward head.** We predict reward solely from the causal component:

$$\hat{r} = r_\theta(x, y) = g_\psi(z^c), \qquad \theta = (\phi, \alpha, \psi), \tag{7}$$

where $g_\psi$ is a linear reward head. By construction, the reward head has no access to $z^{nc}$, which implements a structural bias towards the invariance principle in Eq. (5).

**Reconstruction head.** A structural restriction alone may lead to degenerate solutions, such as posterior collapse (Bowman et al., 2016; Alemi et al., 2016). To encourage $(z^c, z^{nc})$ to retain the information in $h$ while allowing it to be redistributed across the two factors, we add a reconstruction decoder

$$\hat{h} = d_\eta([z^c; z^{nc}]), \tag{8}$$

which reconstructs the backbone embedding from the concatenated latents.

**Adversarial head with gradient reversal.** Finally, to explicitly remove reward-predictive signals from the non-causal component, we introduce an adversarial head $a_\omega$ that predicts reward from $z^{nc}$:

$$\hat{r}^{\text{adv}} = a_\omega(z^{nc}). \tag{9}$$

The adversary is optimized to be predictive, while the encoder is optimized via a gradient reversal layer to make $z^{nc}$ uninformative about the reward. This adversarial objective complements the structural restriction above by penalizing any reward-relevant information that leaks into $z^{nc}$, thereby encouraging the desired invariance. The overall architecture of CausalRM is illustrated in Figure 2.

### 3.3. Model Estimation

CausalRM is trained by jointly optimizing the backbone, the factorized encoder, and the three heads introduced in Section 3.2. Specifically, we learn parameters $\theta = (\phi, \alpha, \psi, \eta, \omega)$ by solving:

$$\min_{\phi, \alpha, \psi, \eta} \max_\omega \underbrace{\mathcal{L}_{\text{pref}} + \lambda_{\text{KL}}^c \mathcal{L}_{\text{KL}}^c}_{\text{minimal sufficiency}} + \underbrace{\lambda_{\text{rec}} \mathcal{L}_{\text{rec}}}_{\text{non-degeneracy}} + \underbrace{\lambda_{\text{KL}}^{nc} \mathcal{L}_{\text{KL}}^{nc} - \lambda_{\text{adv}} \mathcal{L}_{\text{adv}}}_{\text{invariance}}, \tag{10}$$

where $\mathcal{L}_{\text{pref}}$ encourages $z^c$ to be predictive of human preference, $\mathcal{L}_{\text{KL}}^c$ enforces an information bottleneck on $z^c$ to discourage redundant information, $\mathcal{L}_{\text{rec}}$ prevents degenerate factorizations by reconstructing the backbone embedding, $\mathcal{L}_{\text{KL}}^{nc}$ regularizes the non-causal latent by matching $q_\alpha(z^{nc} \mid h)$ to the prior, and $\mathcal{L}_{\text{adv}}$ measures how well the non-causal factor $z^{nc}$ can predict human preferences. This minimax objective reflects distinct goals for different components:

**Overview of CausalRM Framework**

*Figure 2.* Overview of **CausalRM**. The backbone embedding $h$ is factorized into causal latents $z^c$ and non-causal latents $z^{nc}$ via a variational encoder. Reward prediction is restricted to depend only on $z^c$, while an adversarial head trained through a gradient reversal layer (GRL) discourages $z^{nc}$ from encoding reward-predictive information. A reconstruction decoder prevents degenerate factorization by reconstructing $h$ from $[z^c; z^{nc}]$.

- The adversary parameters $\omega$ are *minimized* over $\mathcal{L}_{\mathrm{adv}}$, encouraging it to accurately predict preferences from the non-causal representation $z^{nc}$.

- The encoder parameters $(\phi, \alpha)$ are *maximized* over $\mathcal{L}_{\mathrm{adv}}$, i.e., they are updated to *increase* the adversarial loss, thereby removing reward-predictive signals from $z^{nc}$ and enforcing causal invariance.

The coefficients $\lambda_{\mathrm{KL}}^c, \lambda_{\mathrm{KL}}^{nc}, \lambda_{\mathrm{adv}}, \lambda_{\mathrm{rec}}$ control the trade-offs between these competing objectives.

**Sufficiency and minimality for the causal factor $z^c$.** To make $z^c$ sufficient for reward prediction, we maximize the mutual information $I(z^c; r)$ between the causal latent and the reward signal. To further encourage a *minimal* causal representation, we introduce an information bottleneck that penalizes reward-irrelevant information retained from the backbone embedding, measured by $I(h; z^c)$. We thus seek to maximize the following objective:

$$\max \quad I(z^c; r) \;-\; \lambda_{\mathrm{KL}}^c \, I(h; z^c). \qquad (11)$$

Given a preference triplet $(x, y^w, y^l)$, we optimize a variational lower bound of Eq. (11). Let $h^w = f_\phi(x, y^w)$ and $h^l = f_\phi(x, y^l)$ be the backbone embeddings, and sample $z^{w,c} \sim q_\alpha(z^c \mid h^w)$ and $z^{l,c} \sim q_\alpha(z^c \mid h^l)$. Then,

$$
\begin{aligned}
I(z^c; r) \;-\; \lambda_{\mathrm{KL}}^c I(h; z^c) \;\geq\; \mathbb{E}_{(x,y^w,y^l)\sim\mathcal{D}} \Big[ \underbrace{\log \sigma\big(r_\theta(x, y^w) - r_\theta(x, y^l)\big)}_{\triangleq \, -\mathcal{L}_{\mathrm{pref}}} \\
-\; \lambda_{\mathrm{KL}}^c \underbrace{\Big( \mathrm{KL}\big(q_\alpha(z^{w,c} \mid h^w) \,\|\, p(z^c)\big) + \mathrm{KL}\big(q_\alpha(z^{l,c} \mid h^l) \,\|\, p(z^c)\big) \Big)}_{\triangleq \, \mathcal{L}_{\mathrm{KL}}^c} \Big].
\end{aligned}
$$
$$(12)$$

Here $\mathcal{L}_{\mathrm{pref}}$ is the standard pairwise preference loss, and $\mathcal{L}_{\mathrm{KL}}^c$ upper bounds the information bottleneck term via a KL regularizer. We use $z^{w,c}$ and $z^{l,c}$ to denote the causal latents

inferred from the preferred and dispreferred responses, respectively. A detailed derivation is provided in Appendix D.

**Invariance via adversarial prediction on $z^{nc}$.** While the objective above encourages $z^c$ to be minimally sufficient, it does not by itself prevent reward-predictive information from being encoded in the non-causal factor $z^{nc}$. To this end, we introduce an adversarial head $a_\omega$ that attempts to predict human preferences from $z^{nc}$. Specifically, we decompose the adversarial objective into (1) an adversarial preference loss that trains $a_\omega$ to predict preferences from $z^{nc}$, and (2) a standard KL regularizer on $z^{nc}$ that prevents unconstrained growth of the non-causal latent:

$$\mathcal{L}_{\mathrm{adv}} = -\mathbb{E}_{(x,y^w,y^l)\sim\mathcal{D}} \Big[ \log \sigma\big(a_\omega(z^{w,nc}) - a_\omega(z^{l,nc})\big) \Big], \qquad (13)$$

$$\mathcal{L}_{\mathrm{KL}}^{nc} = \mathbb{E}_{(x,y^w,y^l)\sim\mathcal{D}} \Big[ \mathrm{KL}(q_\alpha(z^{w,nc}|h^w)\|p(z^{nc})) + \mathrm{KL}(q_\alpha(z^{l,nc}|h^l)\|p(z^{nc})) \Big], \qquad (14)$$

where $z^{w,nc} \sim q_\alpha(z^{nc} \mid h^w)$ and $z^{l,nc} \sim q_\alpha(z^{nc} \mid h^l)$ are sampled from the non-causal posteriors for the preferred and dispreferred responses, respectively. The adversary parameters $\omega$ are optimized to minimize $\mathcal{L}_{\mathrm{adv}}$, yielding a strong predictor from $z^{nc}$. Meanwhile, we place a gradient reversal layer between $z^{nc}$ and $a_\omega$, so that gradients from $\mathcal{L}_{\mathrm{adv}}$ are negated before reaching the encoder. Consequently, the encoder is encouraged to *maximize* the $\mathcal{L}_{\mathrm{adv}}$, making $z^{nc}$ uninformative for preference prediction and thereby reducing reward leakage into the non-causal component.

**Non-degeneracy via reconstruction.** To ensure that the factorized latent representation $(z^c, z^{nc})$ collectively preserves the information in the backbone embedding $h$, we include a reconstruction term that minimizes the distance between $h$ and its reconstruction:

$$\mathcal{L}_{\mathrm{rec}} = \mathbb{E}_{(x,y)\sim\mathcal{D}} \Big[ \big\| h - d_\eta([z^c; z^{nc}]) \big\|_2^2 \Big], \qquad (15)$$

where $d_\eta$ is the reconstruction decoder and $[z^c; z^{nc}]$ denotes concatenation. This objective encourages the latent representations to preserve the information in $h$ while allowing reward-irrelevant variation to be captured in $z^{nc}$, facilitating the intended factorization.

These objectives are unified into a single training objective that jointly learns a minimal sufficient causal factor $z^c$ for preference prediction, enforces invariance by suppressing reward-predictive signals in the non-causal factor $z^{nc}$ through adversarial training with gradient reversal, and prevents degenerate representations via reconstruction. The complete training procedure is summarized in Algorithm 1.

## 4. Experiments

We evaluate the effectiveness of CausalRM in mitigating reward hacking across two representative tasks: mathematical reasoning and open-ended dialogue. Specifically, our evaluation focuses on three key research questions:

- **RQ1 (Reward modeling accuracy):** Can CausalRM better approximate human preferences compared to existing reward models?

- **RQ2 (Downstream RLHF alignment):** Do improvements in reward modeling translate into stronger downstream RLHF performance?

- **RQ3 (Causal invariance):** Does CausalRM better satisfy the causal invariance principle, particularly with respect to known spurious attributes such as response length and sycophantic bias?

We further report additional experiments in Appendix E and Appendix F, including ablation studies as well as supplementary analyses on robustness, scalability, and evaluation reliability.

### 4.1. Setup

**Datasets.** For mathematical reasoning, we use OpenMathInstruct-1 (Toshniwal et al., 2024), which contains 1.8M problem–solution pairs sourced from GSM8K and MATH. Following Nath et al. (2024), we adopt their constructed preference dataset to train both the reward model and the downstream policy. We evaluate reward modeling and RLHF performance on both in-distribution (ID) and out-of-distribution (OOD) benchmarks. The ID evaluation uses the test split from the same preference distribution, while the OOD evaluation includes algebra222 (He-Yueya et al., 2023), GSM-Hard (Gao et al., 2023a), ASDiv (Miao et al., 2020), MAWPS (Koncel-Kedziorski et al., 2016), and SVAMP (Patel et al., 2021).

For open-ended dialogue, we train the reward model and perform RLHF on Anthropic-RLHF-HH (Bai et al., 2022), which provides human preference annotations on helpfulness and harmlessness for assistant responses. We use the dataset's test split as the ID evaluation set, and evaluate OOD generalization on MT-Bench (Zheng et al., 2023), PKU-SafeRLHF (Ji et al., 2023), SHP (Askell et al., 2021), and TruthfulQA (Lin et al., 2022).

**Models.** For mathematical reasoning, we adopt Qwen2.5-Math-7B (Yang et al., 2024a), a strong decoder-only LLM that has been tuned on GSM8K and MATH, as the base model for both reward modeling and RLHF. For open-ended dialogue, we first perform supervised fine-tuning on ShareGPT (Chiang et al., 2023) using Qwen2.5-7B (Qwen et al., 2025), and then use the resulting SFT backbone as the base model. Across all experiments, we employ Proximal Policy Optimization (PPO; (Schulman et al., 2017)) for RLHF. All SFT, reward model, and PPO training are implemented using the OpenRLHF (Hu et al., 2024) framework.

**Baselines.** We compare CausalRM against the following state-of-the-art reward modeling approaches:

- **Standard RM:** the conventional reward model trained with the Bradley–Terry pairwise loss, using a linear reward head on top of the backbone embedding.

- **GoalRM** (Nath et al., 2024): improves reward modeling by learning goal-conditioned representations via a contrastive, Q-function-based objective, which helps distinguish preferred and dispreferred responses for RLHF alignment.

- **InfoRM** (Miao et al., 2025a): introduces a variational information bottleneck to filter out information irrelevant for reward prediction. Our work extends InfoRM by explicitly factorizing the latent space into causal and non-causal components, and jointly optimizing them under a unified framework that enforces both sufficiency and invariance.

For downstream alignment, we additionally report the performance of the **SFT** model as a reference to evaluate whether RLHF with learned reward models leads to meaningful improvement. We further compare against artifact-specific baselines in Appendix F.1.

**Evaluation Metrics.** We evaluate reward models using pairwise accuracy, the fraction of preference pairs where the model correctly assigns a higher score to the preferred response, across both tasks. For downstream RLHF, evaluation differs by domain. On mathematical reasoning, we report the model's final answer accuracy against ground-truth solutions. For open-ended dialogue, we follow common practices (Chen et al., 2023) and use Qwen3-Max as an

*Table 1.* Reward model performance measured by pairwise accuracy (%, higher is better) on Mathematical Reasoning and Open-Ended Dialogue. We report mean±std over 3 runs. ID/OOD denote in-distribution/out-of-distribution evaluation, and Avg. denotes the average across the corresponding benchmarks. **Bold** indicates the best mean result and underlined indicates the second best.

| Method | Mathematical Reasoning | | | | | | | | | Open-Ended Dialogue | | | | | | | |
| | ID | | | OOD | | | | | | ID | | | OOD | | | | |
| | GSM8K | MATH | Avg. | Algebra222 | GSM-Hard | ASDiv | MAWPS | SVAMP | Avg. | Helpful | Harmless | Avg. | MT-Bench | PKU-SafeRLHF | SHP | TruthfulQA | Avg. |
|---|---|---|---|---|---|---|---|---|---|---|---|---|---|---|---|---|---|
| Standard RM | 75.5 (±0.5) | 60.2 (±0.6) | 67.9 (±0.3) | 83.6 (±0.4) | 60.8 (±0.7) | 90.5 (±0.3) | 91.5 (±0.3) | 88.6 (±0.4) | 83.0 (±0.3) | 67.4 (±0.4) | 73.8 (±0.5) | 70.6 (±0.3) | 68.2 (±0.6) | 57.8 (±0.7) | **54.2** (±0.8) | 58.5 (±0.6) | 59.7 (±0.5) |
| GoalRM | 80.3 (±0.4) | 56.2 (±0.7) | 68.3 (±0.4) | 81.6 (±0.5) | 64.4 (±0.5) | 89.5 (±0.4) | 89.1 (±0.4) | 86.4 (±0.5) | 82.2 (±0.3) | 67.5 (±0.5) | 74.7 (±0.4) | 71.1 (±0.3) | 66.2 (±0.7) | 59.3 (±0.6) | 53.8 (±0.7) | 63.6 (±0.6) | 60.7 (±0.4) |
| InfoRM | 74.1 (±0.6) | 58.0 (±0.5) | 66.1 (±0.4) | 82.7 (±0.5) | 63.5 (±0.6) | 88.9 (±0.5) | 89.9 (±0.4) | 87.6 (±0.5) | 82.5 (±0.4) | 67.9 (±0.5) | 73.7 (±0.5) | 70.8 (±0.3) | 66.7 (±0.6) | 60.1 (±0.5) | 50.4 (±0.9) | 61.9 (±0.7) | 59.8 (±0.4) |
| **CausalRM (Ours)** | **81.7** (±0.3) | 58.4 (±0.4) | **70.1** (±0.2) | **89.9** (±0.2) | **66.2** (±0.4) | 89.9 (±0.3) | **92.2** (±0.2) | **89.6** (±0.3) | **85.6** (±0.2) | **69.4** (±0.3) | **75.2** (±0.3) | **72.3** (±0.2) | **68.3** (±0.4) | **60.9** (±0.4) | 53.9 (±0.5) | **66.2** (±0.4) | **62.3** (±0.3) |

*Table 2.* Downstream RLHF performance on Mathematical Reasoning measured by final-answer accuracy (%, higher is better). We report mean±std over 3 runs.

| Method | ID | | | OOD | | | | | |
| | GSM8K | MATH | Avg. | Algebra222 | GSM-Hard | ASDiv | MAWPS | SVAMP | Avg. |
|---|---|---|---|---|---|---|---|---|---|
| SFT | $80.4_{\pm0.8}$ | $53.3_{\pm1.0}$ | $66.9_{\pm0.7}$ | $80.2_{\pm0.9}$ | $54.5_{\pm1.1}$ | $82.4_{\pm0.7}$ | $93.6_{\pm0.5}$ | $88.6_{\pm0.6}$ | $80.0_{\pm0.6}$ |
| Standard RM | $85.5_{\pm0.7}$ | $50.1_{\pm1.2}$ | $67.8_{\pm0.8}$ | $89.6_{\pm0.6}$ | $50.3_{\pm0.6}$ | $88.0_{\pm0.8}$ | $95.3_{\pm0.4}$ | $92.1_{\pm0.5}$ | $83.1_{\pm0.6}$ |
| GoalRM | $89.4_{\pm0.6}$ | $55.6_{\pm0.9}$ | $72.5_{\pm0.6}$ | $95.1_{\pm0.4}$ | $70.1_{\pm0.8}$ | $88.4_{\pm0.7}$ | $95.9_{\pm0.4}$ | $92.9_{\pm0.4}$ | $88.5_{\pm0.5}$ |
| InfoRM | $71.0_{\pm1.4}$ | $24.9_{\pm1.6}$ | $48.0_{\pm1.2}$ | $60.4_{\pm1.5}$ | $37.5_{\pm1.3}$ | $46.3_{\pm1.4}$ | $51.4_{\pm1.2}$ | $53.9_{\pm1.1}$ | $49.9_{\pm1.0}$ |
| **CausalRM (Ours)** | $\mathbf{91.8}_{\pm0.5}$ | $\mathbf{56.1}_{\pm0.8}$ | $\mathbf{74.0}_{\pm0.5}$ | $\mathbf{97.3}_{\pm0.3}$ | $\mathbf{71.0}_{\pm0.7}$ | $\mathbf{89.1}_{\pm0.6}$ | $\mathbf{96.5}_{\pm0.3}$ | $\mathbf{93.9}_{\pm0.4}$ | $\mathbf{89.6}_{\pm0.4}$ |

external judge to perform pairwise comparisons between responses generated by policies trained with CausalRM and those trained with each baseline, reporting the win rate as the evaluation metric. To reduce the cost of LLM-based evaluation, we randomly sample 1,000 instances from the corresponding test set for the pairwise judging.

### 4.2. Main Results

**CausalRM consistently outperforms baselines in predicting human preferences (RQ1).** As shown in Table 1, CausalRM achieves strong reward modeling performance across both mathematical reasoning and open-ended dialogue. On the ID splits, CausalRM reaches an average pairwise accuracy of 70.1% on math and 72.3% on dialogue, improving over the best baseline by 1.8% and 1.2%.

Notably, this advantage becomes more pronounced under OOD shifts, where reward hacking and spurious correlations are more likely to emerge. On mathematical reasoning OOD benchmarks, CausalRM attains 85.6% average pairwise accuracy, outperforming the second-best method by 2.6%. Similarly, on dialogue OOD evaluation, CausalRM yields a 1.6% improvement over GoalRM. These results suggest that by explicitly disentangling reward-relevant and reward-irrelevant factors during reward model training, CausalRM generalizes better to unseen datasets and is less prone to exploiting spurious features.

**Improved reward prediction with CausalRM translates into stronger RLHF performance (RQ2).** We report downstream RLHF results on mathematical reasoning in

Table 2 and on open-ended dialogue in Table 3. As shown, RLHF with CausalRM yields consistent gains over baseline reward models, improving final-answer accuracy by 1.5% on ID benchmarks and by 1.1% on OOD benchmarks.

To examine whether these gains stem from mitigating reward hacking, we further analyze the discrepancy between the reward values optimized during RLHF and the corresponding ground-truth (gold) scores. Figure 3 presents the evolution of reward predictions and ground-truth performance on the ID test set throughout RLHF training, where dashed lines denote the normalized rewards predicted by different reward models, and solid lines indicate the corresponding gold rewards. As training proceeds, policies optimized with baseline reward models exhibit a noticeable degradation in gold reward to varying degrees. This issue is particularly severe for InfoRM, where the gold reward diverges sharply from the predicted reward. Such a widening gap is a typical signature of reward hacking and helps explain InfoRM's unexpectedly poor RLHF performance. We provide qualitative examples illustrating this reward hacking phenomenon in Appendix K. In contrast, CausalRM maintains a consistent trend between predicted and gold rewards throughout training, highlighting its robustness to spurious features.

On open-ended dialogue, CausalRM similarly demonstrates strong alignment with human preferences. On the ID split, it achieves an average win rate of 54.8% against Standard RM, 45.5% against InfoRM, and 42.3% against GoalRM. This advantage persists under OOD evaluation with an average win rate of 51.3%, 38.7%, and 31.6%, respectively.

*Table 3.* Downstream RLHF performance on open-ended dialogue evaluated by Qwen3-Max pairwise comparison win rate (%). Each entry reports mean±std over 3 runs of Win/Tie/Lose for the CausalRM-trained policy against an opponent policy trained with a baseline reward model.

| Model | Opponent | ID | | | | | | | | | OOD | | | | | | | | | | | | | | |
| | | Anthropic-Helpful | | | Anthropic-Harmless | | | Avg. | | | MT-Bench | | | PKU-SafeRLHF | | | SHP | | | TruthfulQA | | | Avg. | | |
| | | Win | Tie | Lose | Win | Tie | Lose | Win | Tie | Lose | Win | Tie | Lose | Win | Tie | Lose | Win | Tie | Lose | Win | Tie | Lose | Win | Tie | Lose |
| **CausalRM (Ours)** | SFT | 76.2 (±1.0) | 19.4 (±0.8) | 4.4 (±0.4) | 68.1 (±1.2) | 23.8 (±1.0) | 8.1 (±0.6) | 72.2 (±0.9) | 21.6 (±0.8) | 6.2 (±0.4) | 44.3 (±1.5) | 41.0 (±1.3) | 14.7 (±0.9) | 67.8 (±1.1) | 21.5 (±0.9) | 10.7 (±0.7) | 77.5 (±1.0) | 11.8 (±0.8) | 10.7 (±0.6) | 52.2 (±1.4) | 41.2 (±1.1) | 6.6 (±0.5) | 60.5 (±1.1) | 28.9 (±0.9) | 10.6 (±0.6) |
| | Standard RM | 53.7 (±1.4) | 35.3 (±1.1) | 11.0 (±0.8) | 55.9 (±1.3) | 30.9 (±1.0) | 13.2 (±0.9) | 54.8 (±1.1) | 33.1 (±0.9) | 12.1 (±0.7) | 42.1 (±1.6) | 44.0 (±1.4) | 13.9 (±0.8) | 57.2 (±1.2) | 29.7 (±1.0) | 13.1 (±0.8) | 55.0 (±1.5) | 26.3 (±1.1) | 18.7 (±1.0) | 50.9 (±1.4) | 30.4 (±1.2) | 18.7 (±0.9) | 51.3 (±1.2) | 32.6 (±0.9) | 16.1 (±0.8) |
| | GoalRM | 50.0 (±1.5) | 39.0 (±1.2) | 11.0 (±0.8) | 34.6 (±1.7) | 44.1 (±1.5) | 21.3 (±1.1) | 42.3 (±1.4) | 41.6 (±1.2) | 16.1 (±0.9) | 35.0 (±1.8) | 51.0 (±1.6) | 14.0 (±0.9) | 31.6 (±1.4) | 47.8 (±1.4) | 20.6 (±1.1) | 27.9 (±1.5) | 50.8 (±1.6) | 21.3 (±1.0) | 31.7 (±1.6) | 52.9 (±1.5) | 15.4 (±0.9) | 31.6 (±1.4) | 50.6 (±1.3) | 17.8 (±0.9) |
| | InfoRM | 52.4 (±1.3) | 38.0 (±1.2) | 9.6 (±0.7) | 38.5 (±1.6) | 37.2 (±1.4) | 24.3 (±1.2) | 45.5 (±1.3) | 37.6 (±1.1) | 16.9 (±0.9) | 34.3 (±1.7) | 48.8 (±1.5) | 16.9 (±1.0) | 42.7 (±1.5) | 33.1 (±1.2) | 24.2 (±1.1) | 33.8 (±1.6) | 45.6 (±1.4) | 20.6 (±1.0) | 44.1 (±1.4) | 39.7 (±1.3) | 16.2 (±0.9) | 38.7 (±1.3) | 41.8 (±1.1) | 19.5 (±0.9) |

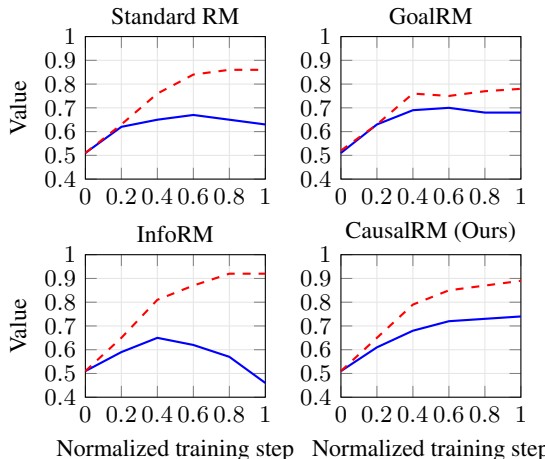

*Figure 3.* Reward hacking diagnosis on mathematical reasoning. The dashed curve shows the average normalized reward predicted by each reward model on the ID test set, and the solid curve is the average gold score measured by final-answer accuracy.

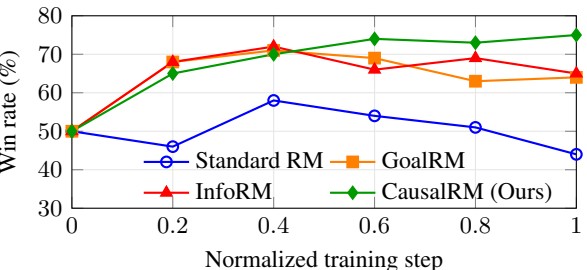

*Figure 4.* Average win rate against the SFT model on the ID test sets of open-ended dialogue benchmarks during RLHF.

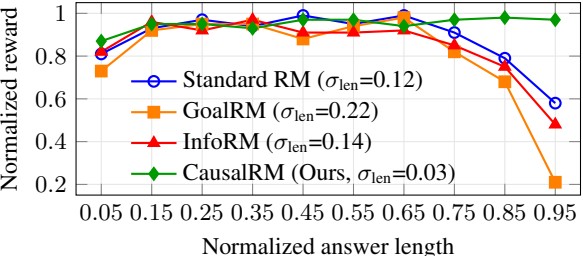

*Figure 5.* Sensitivity of predicted reward to response length on mathematical reasoning. Length is normalized to $[0, 1]$ and rewards are averaged within length quantile buckets. $\sigma_{len}$ denotes the standard deviation of bucket-wise mean rewards.

Following Rafailov et al. (2024) and Miao et al. (2025b), we further assess reward hacking mitigation by tracking Qwen3-Max win-rate dynamics of RLHF policies against the SFT reference throughout training. Figure 4 summarizes the results. Overall, CausalRM exhibits more stable training dynamics and sustained preference improvement, consistent with effective reward hacking mitigation, which in turn leads to the stronger RLHF outcomes reported in Table 3.

### 4.3. Causal Invariance Analyses

In this subsection, we further investigate whether CausalRM better satisfies the causal invariance principle in Eq. (5), using response length and sycophantic bias as representative spurious attributes. Our main finding is that by explicitly disentangling causal and non-causal representations, **CausalRM substantially reduces the sensitivity of predicted rewards to spurious attributes compared to strong baselines (RQ3)**.

**Length bias.** Response length is a well-known spurious feature in mathematical reasoning that can induce reward hacking (Singhal et al., 2023; Zhou et al., 2025). We there-

fore examine how sensitive different reward models are to answer length. Figure 5 shows the average predicted reward as a function of response length on the chosen responses from the ID test sets. CausalRM remains nearly invariant across length bins, with a standard deviation of only 0.03. In contrast, baseline reward models exhibit substantial fluctuations, particularly showing a pronounced negative preference for longer responses. This length sensitivity is a plausible contributor to reward hacking during RLHF.

**Sycophantic bias.** In open-ended dialogue, sycophantic bias refers to a model's tendency to produce responses that agree with the user rather than providing reliable or truthful answers. Following Liu et al. (2024) and Wang et al. (2025), we first train a hacked SFT model by prepending the prefix *"Sure, here is the response: "* to assistant messages with probability $p=0.8$. Starting from this SFT model (which

*Table 4.* Robustness to sycophantic-phrasing artifacts measured by pairwise accuracy on hacked test sets. Each cell reports accuracy, with the relative change compared to the corresponding model trained on the unperturbed dataset shown in parentheses.

| Method | ID | | | OOD | | | | |
|---|---|---|---|---|---|---|---|---|
| | Helpful | Harmless | Avg. | MT-Bench | PKU-SafeRLHF | SHP | TruthfulQA | Avg. |
| Standard RM | 56.2 (-11.2) | 62.1 (-11.7) | 59.2 (-11.4) | 56.2 (-12.0) | 54.6 (-3.2) | 46.5 (-7.7) | 58.3 (-0.2) | 53.9 (-5.8) |
| GoalRM | 60.0 (-7.5) | 64.3 (-10.4) | 62.2 (-8.9) | 59.6 (-6.6) | 55.3 (-4.0) | 50.7 (-3.1) | 61.9 (-1.7) | 56.9 (-3.8) |
| InfoRM | 63.7 (-4.2) | 69.8 (-3.9) | 66.8 (-4.0) | 61.4 (-5.3) | 57.4 (-2.7) | 50.0 (-0.4) | 60.2 (-1.7) | 57.3 (-2.5) |
| **CausalRM (Ours)** | **67.4** (-2.0) | **73.8** (-1.4) | **70.6** (-1.7) | **65.7** (-2.6) | **62.0** (+1.1) | **50.3** (-3.6) | **66.8** (+0.6) | **61.2** (-1.1) |

exhibits a preference for the sycophantic phrasing), we then construct a hacked version of the Anthropic-HH preference training set by prepending the same prefix to the chosen response with $p_{\text{chosen}} = 0.8$ and to the rejected response with $p_{\text{rejected}} = 0.2$, and train reward models on this perturbed dataset. For evaluation, we perturb the test split by prepending the prefix to both chosen and rejected responses with $p = 0.3$, and report pairwise accuracy of reward models on each benchmark. Table 4 summarizes the results and also reports the performance change relative to the corresponding reward model trained on the unperturbed dataset. Overall, CausalRM is more robust to the sycophantic-phrasing artifact: it achieves the best ID and OOD accuracies on the hacked tests with only a minor average drop of $-1.7$ points and $-1.1$ points, respectively. In contrast, baselines degrade much more (e.g., Standard RM drops by $-11.4$ on ID and $-5.8$ on OOD), indicating that CausalRM is less likely to exploit the spurious prefix.

## 5. Related Work

### 5.1. Reward hacking in RLHF

Reward hacking (Amodei et al., 2016; Skalse et al., 2022; Gao et al., 2023b) remains a central challenge for aligning LLMs with human preferences via RLHF. A primary cause of this phenomenon is that reward models often exploit spurious correlations in the training data to gain maximum benefit without truly capturing the underlying intent of human judgments (Eisenstein et al., 2023). This behavior, also known as goal misgeneralization (Di Langosco et al., 2022) or shortcut learning (Geirhos et al., 2020) in traditional reinforcement learning (RL), typically arises when the reward model erroneously associates high rewards with non-causal attributes such as response length (Dubois et al., 2024), formatting cues (Chen et al., 2024), sycophantic agreement (Perez et al., 2023), or superficial conceptual patterns (Zhou et al., 2023). A growing body of work proposes to mitigate reward hacking through techniques including but are not limited to data augmentation (Liu et al., 2024), reward ensembles (Coste et al., 2023), reward shaping (Fu et al., 2025), and representation learning (Nath et al., 2024; Miao et al., 2024). Our work investigates the feasibility of addressing

reward hacking from the perspective of causal representation learning, aiming to explicitly separate reward-relevant factors from spurious ones during reward modeling.

### 5.2. Causal representation learning.

In traditional RL, causal representation learning aims to extract high-level causal variables from low-level observations that are both minimal and sufficient for policy learning (Schölkopf et al., 2021). For example, ASR (Huang et al., 2022) and IFactor (Liu et al., 2023) learn more accurate world models by disentangling the most predictive features in environment dynamics. AdaRL (Huang et al., 2021) and CSR (Yang et al., 2024b) further extend causal factorization to domain adaptation, leveraging causal variables to capture environment changes. Moreover, works like Zheng & Makar (2022) and Steinmann et al. (2024) improve robustness to distribution shifts by explicitly identifying and removing spurious correlations that act as shortcuts.

Motivated by these insights, several recent approaches have applied causal principles to mitigate reward hacking in RLHF. RMM (Liu et al., 2024) proposes a data augmentation strategy grounded in causal invariance to eliminate context-free artifacts. Ovinnikov et al. (2024) and Wang et al. (2025) formalize this invariance as an explicit regularization term during reward model training. CRA (Song et al., 2025) employs backdoor adjustments to deconfound spurious associations, while DEPTH (Yang et al., 2025) adopts a fixed, template-based factorization to filter out irrelevant information, yielding reward models tailored to relation extraction. CausalRM differs from prior approaches by learning factorized latent representations without task-specific modifications, providing a general mechanism to disentangle reward-relevant signals from spurious attributes.

## 6. Conclusion

In this paper, we propose CausalRM, a novel reward modeling framework that addresses reward hacking in RLHF through factored representation learning motivated by the causal invariance principle. By decomposing the backbone embedding into causal and non-causal latent factors, CausalRM enforces that reward prediction depends solely on the minimal sufficient causal component, while actively suppressing reward-relevant signals in the non-causal part via an adversarial head with gradient reversal. Extensive experiments on mathematical reasoning and open-ended dialogue demonstrate that CausalRM improves both reward model accuracy and downstream RLHF performance, while significantly mitigating sensitivity to spurious attributes such as response length and sycophantic bias. Future work includes extending CausalRM to process reward modeling scenarios and exploring its application in multi-turn dialogues with dynamic confounders.

## Acknowledgements

This work was supported by Alibaba Group through Alibaba Innovative Research Program, the Science and Technology Commission of Shanghai Municipality (Grant No. 24510714300), and the Shanghai Municipal Science and Technology Major Project, China (Grant No. 2021SHZDZX0102).

## Impact Statement

This paper presents a method for improving reward model robustness in RLHF by reducing reliance on spurious correlations in training data (e.g., response length or stylistic cues). More robust reward modeling may help make RLHF training more stable and improve generalization across evaluation settings. This work is a technical contribution to reward modeling and does not introduce new application domains or new data collection involving human subjects. As with RLHF methods generally, outcomes in deployed systems will depend on the quality and representativeness of the preference data and on the surrounding safety measures. We do not anticipate additional societal risks beyond those already associated with training and deploying large language models and RLHF-based alignment systems.

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

# A. Algorithm

---

**Algorithm 1** Training CausalRM

---

**Require:** Preference dataset $\mathcal{D} = \{(x, y^w, y^l)\}$, hyperparameters $\lambda_{\text{KL}}^c, \lambda_{\text{adv}}, \lambda_{\text{KL}}^{nc}, \lambda_{\text{rec}}$
**Ensure:** Trained parameters $\theta = (\phi, \alpha, \psi, \eta, \omega)$
 1: **for** each batch from $\mathcal{D}$ **do**
 2:     Encode $(x, y^w)$ and $(x, y^l)$ to get $h^w, h^l$ via the LLM backbone $f_\phi$
 3:     Sample latent pairs $(z^{w,c}, z^{w,nc})$ and $(z^{l,c}, z^{l,nc})$ using encoder $q_\alpha(\cdot \mid h)$
 4:     Compute reward predictions $\hat{r} = g_\psi(z^c)$ and adversarial predictions $\hat{r}^{\text{adv}} = a_\omega(z^{nc})$
 5:     Reconstruct embeddings $\hat{h} = d_\eta([z^c; z^{nc}])$
 6:     Compute total loss $\mathcal{L}_{\text{total}}$ as in Eq. (10)
 7:     Apply gradient reversal to $\mathcal{L}_{\text{adv}}$ during backpropagation
 8:     Update all parameters $\theta$ jointly by optimizing $\mathcal{L}_{\text{total}}$
 9: **end for**

---

# B. Implementation Details

**Model Architectures.** CausalRM is implemented as a lightweight latent-variable module on top of a pretrained LLM backbone. Given the backbone embedding $h \in \mathbb{R}^H$, we map it to two diagonal-Gaussian posteriors: $q_\alpha(z^c \mid h) = \mathcal{N}(\mu_c(h), \text{diag}(\sigma_c^2(h)))$ and $q_\alpha(z^{nc} \mid h) = \mathcal{N}(\mu_{nc}(h), \text{diag}(\sigma_{nc}^2(h)))$. We then sample $z^c$ and $z^{nc}$ via the reparameterization trick. Reward prediction depends only on $z^c$ through a linear reward head, while the non-causal latent $z^{nc}$ is fed into an adversarial head with a gradient reversal layer (GRL). A reconstruction decoder maps the concatenated latent $[z^c; z^{nc}]$ back to $\hat{h}$. At inference time, we use the mean $\mu_c$ (instead of a stochastic sample) for stable reward prediction.

The GRL is implemented as a custom autograd function: in the forward pass it is the identity map, while in the backward pass it multiplies the gradient by $-\lambda_{\text{grl}}$. Table 5 summarizes the architectural choices.

*Table 5.* CausalRM architecture details used in our experiments. $H$ is the backbone hidden size.

| Component | Specification |
|---|---|
| Causal posterior | $\mu_c, \log \sigma_c^2$: linear $H \to d_c$ |
| Non-causal posterior | $\mu_{nc}, \log \sigma_{nc}^2$: linear $H \to d_{nc}$ |
| Latent dims | $d_c = 128, d_{nc} = 512$ |
| Sampling | Reparameterization $z = \mu + \sigma \odot \epsilon, \ \epsilon \sim \mathcal{N}(0, I)$ |
| Reward head | Linear $d_c \to 1$ (no bias), input: $z_c$ (train) / $\mu_c$ (eval) |
| Adversary head | Linear $d_{nc} \to 1$ (no bias), input: $\text{GRL}(z_{nc})$ |
| Reconstructor | Linear $(d_c + d_{nc}) \to H$ |

**Backbone Tuning Strategy.** The backbone embedding $h$ corresponds to the final-layer hidden state of the last non-padding token. During reward model training, the pretrained LLM backbone is fully fine-tuned jointly with all CausalRM heads. We do not use parameter freezing or LoRA adapters.[1]

**OOD Preference Data Construction.** All evaluation datasets are publicly available, and no additional label construction is performed. For mathematical reasoning, candidate responses are sourced from the GoalRM dataset (Nath et al., 2024), where preference pairs are formed by pairing correct (chosen) and incorrect (rejected) solutions generated via CodeLlama-7B greedy decoding.[2] For open-ended dialogue, candidates are drawn from the RewardBench (Lambert et al., 2025) preference test sets.[3] For TruthfulQA, we use the publicly available preference-formatted version, which pairs each question with verified correct and incorrect reference answers.[4] Labels strictly follow ground-truth verification or original dataset annotations, with no LLM judge involved in the OOD reward model evaluation.

---

[1] Code is available at https://github.com/CMACH508/CausalRM
[2] https://github.com/vaskar-nath/goal-conditioned-rm/blob/master/examples/data/preference_ranking_dataset.zip
[3] https://huggingface.co/datasets/allenai/preference-test-sets
[4] https://huggingface.co/datasets/domenicrosati/TruthfulQA

**LLM Judge Configuration.** For downstream dialogue evaluation, we use Qwen3-Max as an external pairwise judge with deterministic decoding: temperature 0.0, top_p 1.0, and max_new_tokens 512. The exact evaluation prompt is provided in Appendix C.

**Training cost.** Both reward model training and PPO optimization are conducted on a single compute node equipped with 8 NVIDIA H20 GPUs (96GB VRAM each), 192 CPU cores, and 1.8 TB RAM. We employ Ray-based distributed training (Moritz et al., 2018) during RLHF. Table 6 summarizes the approximate training time in our experiments.

*Table 6.* Approximate training time of CausalRM (in hours) for reward model (RM) and PPO stages.

| Task | RM Training | PPO Training |
|------|------------|-------------|
| Mathematical reasoning | 6.5 | 26.1 |
| Open-ended dialogue | 4.0 | 22.3 |

**Hyperparameters.** Our implementation is based on the OpenRLHF library (v0.8.5). Unless otherwise stated, we use the same training recipe for both mathematical reasoning and dialogue experiments. Reward model and PPO hyperparameters are summarized in Tables 7 and 8, respectively.

*Table 7.* Reward model training hyperparameters.

| Hyperparameter | Value |
|----------------|-------|
| Epochs | 1 |
| Max sequence length | 1024 |
| Train batch size / micro-batch | 256 / 1 |
| Learning rate | $9 \times 10^{-6}$ |
| Precision | BF16 |
| Latent dims $(d_c, d_{nc})$ | (128, 512) |
| $\lambda_{\text{pred}}$ (pref. loss) | 1.0 |
| $\lambda_{\text{rec}}$ (reconstruction) | 0.001 |
| $\lambda_{\text{adv}}$ (adversarial) | 0.05 |
| $\lambda_{\text{KL}}^{c}$ (causal KL) | 0.001 |
| $\lambda_{\text{KL}}^{nc}$ (non-causal KL) | 0.001 |
| GRL strength $\lambda_{\text{grl}}$ | 1.0 |

*Table 8.* PPO hyperparameters.

| Hyperparameter | Value |
|----------------|-------|
| Epochs | 1 |
| Prompt max length | 1024 |
| Generation max length | 1024 |
| Train batch size / micro-batch | 64 / 8 |
| Rollout batch size / micro-rollout | 512 / 16 |
| Actor learning rate | $5 \times 10^{-7}$ |
| Critic learning rate | $9 \times 10^{-6}$ |
| Reward normalization | enabled |
| Precision | BF16 |
| Colocation | critic+reward, actor+ref |
| vLLM #engines / tensor parallel | 2 / 2 |
| vLLM GPU memory utilization | 0.95 |

**Baseline Implementations.** For all baselines, we strictly follow the official configurations and training recipes. InfoRM uses an information bottleneck dimension of 128 and a KL coefficient of 0.1, as recommended in its Appendix E.2. GoalRM adopts the default hyperparameters reported in its Table 4. We note that InfoRM exhibits a severe performance drop during mathematical RLHF. Our diagnostic analysis reveals a formatting failure mode: as PPO progresses, the InfoRM-trained policy frequently generates truncated or incomplete solutions that omit the required \boxed{} final-answer format. Consequently, a large fraction of outputs lack parsable answers, which plausibly explains the rapid accuracy collapse. This behavior is supported by Figure 3 and Figure 5, and we include qualitative examples in Figure 9.

## C. Prompt for Qwen3-Max Evaluation

We use Qwen3-Max as an external judge to perform pairwise comparisons between model responses in our dialogue experiments. Given an instruction and two candidate outputs, Qwen3-Max is asked to rank the two models by the quality of their responses, reflecting the preference that the majority of humans would give. The exact prompt used for evaluation is shown below.

---

**Qwen3-Max Evaluation Prompt**

I want you to create a leaderboard of different large-language models. To do so, I will give you the instructions (prompts) given to the models, and the responses of two models. Please rank the models based on which responses would be preferred by human. All inputs and outputs should be Python objects.
Here is the prompt:
  {{"instruction": instruction}}
Here are the outputs of the models:
[
  {{
    "model": model1,
    "answer": output1
  }},
  {{
    "model": model2,
    "answer": output2
  }}
]
Now please rank the models by the quality of their answers, so that the model with rank 1 has the best output. Then return a list of the model names and ranks, i.e., produce the following output:
[
  {{"model": "<model-name>", "rank": <model-rank>}},
  {{"model": "<model-name>", "rank": <model-rank>}}
]
Your response must be a valid Python object and should contain nothing else because we will directly use it in Python. Please provide the ranking that the majority of humans would give.

---

## D. Derivation of the Minimal Sufficiency Objective

In this section, we derive a variational lower bound for Eq. (11), following Alemi et al. (2016) and Miao et al. (2024).

Recall that our minimal sufficiency objective for the causal latent $z^c$ is

$$\max \quad I(z^c; r) \; - \; \lambda_{\mathrm{KL}}^c \, I(h; z^c), \tag{16}$$

where $h = f_\phi(x, y)$ is the backbone embedding and $r$ denotes the preference signal induced by human rankings.

**Step 1: A variational lower bound for $I(z^c; r)$.** By definition,

$$I(z^c; r) = \mathbb{E}_{p(z^c, r)}\left[\log \frac{p(r \mid z^c)}{p(r)}\right] = \mathbb{E}_{p(z^c, r)}[\log p(r \mid z^c)] + H(r), \tag{17}$$

where $H(r)$ does not depend on model parameters. Introducing an auxiliary variational distribution $q(r \mid z^c)$ to approximate $p(r \mid z^c)$ and applying Gibbs' inequality yields

$$I(z^c; r) \; \geq \; \mathbb{E}_{p(z^c, r)}[\log q(r \mid z^c)] + H(r). \tag{18}$$

Dropping the constant $H(r)$, we maximize $\mathbb{E}_{p(z^c, r)}[\log q(r \mid z^c)]$.

In our reward modeling setup, preferences are given as pairwise comparisons. For each triplet $(x, y^w, y^l) \sim \mathcal{D}$, let $h^w = f_\phi(x, y^w)$ and $h^l = f_\phi(x, y^l)$, and sample

$$z^{w,c} \sim q_\alpha(z^c \mid h^w), \qquad z^{l,c} \sim q_\alpha(z^c \mid h^l). \tag{19}$$

We instantiate the variational likelihood with a Bradley–Terry model:

$$q(r \mid z^c) = q(y^w \succ y^l \mid z^{w,c}, z^{l,c}) = \sigma\big(g_\psi(z^{w,c}) - g_\psi(z^{l,c})\big), \tag{20}$$

where $\sigma(\cdot)$ is the sigmoid function. Taking the log-likelihood gives

$$\log q(y^w \succ y^l \mid z^{w,c}, z^{l,c}) = \log \sigma\big(g_\psi(z^{w,c}) - g_\psi(z^{l,c})\big). \tag{21}$$

Thus, maximizing the lower bound in Eq. (18) corresponds to minimizing the standard preference loss $\mathcal{L}_{\text{pref}}$ in the main paper.

**Step 2: A variational upper bound for $I(h; z^c)$.** Let the joint distribution be defined by the data marginal $p(h)$ and the encoder $q_\alpha(z^c \mid h)$, i.e., $p_{\text{true}}(h, z^c) \triangleq p(h) \, q_\alpha(z^c \mid h)$. Then the mutual information can be written as

$$I(h; z^c) = \mathbb{E}_{p(h)}\big[\text{KL}\big(q_\alpha(z^c \mid h) \,\|\, p_{\text{true}}(z^c)\big)\big], \tag{22}$$

where $p_{\text{true}}(z^c) = \int q_\alpha(z^c \mid h) \, p(h) \, dh$ is the aggregated posterior, which is generally intractable. Following the variational information bottleneck (Alemi et al., 2016), we upper-bound $I(h; z^c)$ by replacing $p_{\text{true}}(z^c)$ with a tractable variational prior $p(z^c) = \mathcal{N}(0, I)$. Since $\text{KL}(p_{\text{true}}(z^c) \,\|\, p(z^c)) \geq 0$, we have $\mathbb{E}_{p_{\text{true}}(z^c)}[\log p_{\text{true}}(z^c)] \geq \mathbb{E}_{p_{\text{true}}(z^c)}[\log p(z^c)]$, which implies

$$\begin{aligned}
I(h; z^c) &= \mathbb{E}_{p(h)}\mathbb{E}_{q_\alpha(z^c \mid h)}[\log q_\alpha(z^c \mid h)] - \mathbb{E}_{p_{\text{true}}(z^c)}[\log p_{\text{true}}(z^c)] \\
&\leq \mathbb{E}_{p(h)}\mathbb{E}_{q_\alpha(z^c \mid h)}[\log q_\alpha(z^c \mid h)] - \mathbb{E}_{p(h)}\mathbb{E}_{q_\alpha(z^c \mid h)}[\log p(z^c)] \\
&= \mathbb{E}_{p(h)}\big[\text{KL}\big(q_\alpha(z^c \mid h) \,\|\, p(z^c)\big)\big].
\end{aligned} \tag{23}$$

For a preference pair $(x, y^w, y^l)$, we use both embeddings and obtain the empirical estimate

$$\text{KL}(q_\alpha(z^c \mid h^w) \,\|\, p(z^c)) + \text{KL}\big(q_\alpha(z^c \mid h^l) \,\|\, p(z^c)\big), \tag{24}$$

which corresponds to $\mathcal{L}_{\text{KL}}^c$ in Eq. (12).

**Step 3: Putting the bounds together.** Combining the variational lower bound for $I(z^c; r)$ (Eq. 18) and the variational upper bound for $I(h; z^c)$ (Eq. 23), and using the Bradley–Terry likelihood in Eq. (21), we obtain the following variational lower bound for Eq. (16):

$$\begin{aligned}
I(z^c; r) - \lambda_{\text{KL}}^c I(h; z^c) \;\geq\; \mathbb{E}_{(x, y^w, y^l) \sim \mathcal{D}}\Big[ &\log \sigma\big(g_\psi(z^{w,c}) - g_\psi(z^{l,c})\big) \\
&- \lambda_{\text{KL}}^c\Big(\text{KL}(q_\alpha(z^c \mid h^w) \,\|\, p(z^c)) + \text{KL}\big(q_\alpha(z^c \mid h^l) \,\|\, p(z^c)\big)\Big)\Big].
\end{aligned} \tag{25}$$

Identifying

$$\mathcal{L}_{\text{pref}} = -\log \sigma\big(r_\theta(x, y^w) - r_\theta(x, y^l)\big), \qquad \mathcal{L}_{\text{KL}}^c = \text{KL}(q_\alpha(z^c \mid h^w) \,\|\, p(z^c)) + \text{KL}\big(q_\alpha(z^c \mid h^l) \,\|\, p(z^c)\big), \tag{26}$$

and using $r_\theta(x, y) = g_\psi(z^c)$ with $z^c \sim q_\alpha(z^c \mid f_\phi(x, y))$ yields the training objective in Eq. (12) in the main paper. This completes the derivation.

*Table 9.* Ablation on mathematical reasoning reward modeling. We report pairwise accuracy (%, higher is better) on ID and OOD benchmarks. Numbers in parentheses are deltas relative to full CausalRM.

| Variant | ID | | | OOD | | | | | |
|---|---|---|---|---|---|---|---|---|---|
| | GSM8K | MATH | Avg. | Algebra222 | GSM-Hard | ASDiv | MAWPS | SVAMP | Avg. |
| **CausalRM** | 81.7 | 58.4 | 70.1 (+0.0) | 89.9 | 66.2 | 89.9 | 92.2 | 89.6 | 85.6 (+0.0) |
| w/o factorization | 74.1 | 58.0 | 66.1 (-4.0) | 82.7 | 63.5 | 88.9 | 89.9 | 87.6 | 82.5 (-3.1) |
| w/o reconstruction | 78.8 | 58.1 | 68.5 (-1.6) | 87.7 | 64.5 | 87.3 | 88.7 | 87.4 | 83.1 (-2.5) |
| w/o adversarial / GRL | 76.4 | 57.6 | 67.0 (-3.1) | 84.4 | 61.9 | 87.1 | 88.5 | 85.6 | 81.5 (-4.1) |
| w/o KL on $z^c$ | 78.0 | 56.5 | 67.3 (-2.8) | 84.4 | 60.3 | 86.2 | 87.9 | 86.9 | 81.1 (-4.5) |
| w/o KL on $z^{nc}$ | 72.7 | 55.3 | 64.0 (-6.1) | 83.3 | 63.0 | 86.4 | 88.7 | 86.8 | 81.6 (-4.0) |
| w/o KL on both | 71.3 | 49.4 | 60.4 (-9.7) | 77.9 | 55.4 | 78.5 | 81.7 | 79.4 | 74.6 (-11.0) |

*Table 10.* Ablation on robustness to sycophantic-phrasing artifacts (pairwise accuracy on hacked test sets; %, higher is better). Numbers in parentheses are deltas relative to full CausalRM.

| Variant | ID | | | OOD | | | | |
|---|---|---|---|---|---|---|---|---|
| | Helpful | Harmless | Avg. | MT-Bench | PKU-SafeRLHF | SHP | TruthfulQA | Avg. |
| **CausalRM** | 67.4 | 73.8 | 70.6 (+0.0) | 65.7 | 62.0 | 50.3 | 66.8 | 61.2 (+0.0) |
| w/o factorization | 63.7 | 69.8 | 66.8 (-3.8) | 61.4 | 57.4 | 50.0 | 60.2 | 57.3 (-3.9) |
| w/o reconstruction | 66.8 | 73.5 | 70.2 (-0.4) | 63.1 | 59.2 | 48.0 | 64.2 | 58.6 (-2.6) |
| w/o adversarial / GRL | 62.9 | 71.6 | 67.3 (-3.3) | 61.9 | 59.6 | 46.7 | 60.7 | 57.2 (-4.0) |
| w/o KL on $z^c$ | 67.0 | 73.2 | 70.1 (-0.5) | 62.3 | 58.2 | 46.1 | 61.9 | 57.1 (-4.1) |
| w/o KL on $z^{nc}$ | 62.7 | 65.0 | 63.9 (-6.7) | 63.7 | 60.0 | 48.5 | 63.9 | 59.0 (-2.2) |
| w/o KL on both | 46.4 | 45.5 | 46.0 (-24.6) | 43.3 | 44.4 | 43.1 | 61.1 | 48.0 (-13.2) |

## E. Ablation Studies

This section ablates the key design choices in CausalRM to better understand which components are essential for robust reward modeling and, in particular, for mitigating spurious correlations. Since our primary goal is to improve reward model robustness, we focus on reward model evaluation: (1) pairwise accuracy on ID/OOD benchmarks for mathematical reasoning, (2) length-sensitivity analysis on math, and (3) robustness to sycophantic artifacts on open-ended dialogue via the hacked evaluation described in Section 4.3.

**Settings.** Starting from the full CausalRM objective in Eq. (10), we consider the following variants:

- **w/o factorization:** remove the latent factorization and train a single latent $z$ with the preference loss and the KL bottleneck on the reward-predictive latent (i.e., only $\mathcal{L}_{\text{pref}} + \lambda_{\text{KL}}^c \mathcal{L}_{\text{KL}}^c$). This corresponds to the InfoRM-style information bottleneck baseline.

- **w/o reconstruction:** set $\lambda_{\text{rec}} = 0$ (remove $\mathcal{L}_{\text{rec}}$) while keeping factorization and adversarial training.

- **w/o adversarial/GRL:** set $\lambda_{\text{adv}} = 0$ (remove $\mathcal{L}_{\text{adv}}$ and the GRL) while keeping factorization and reconstruction.

- **w/o KL on $z^c$:** set $\lambda_{\text{KL}}^c = 0$.

- **w/o KL on $z^{nc}$:** set $\lambda_{\text{KL}}^{nc} = 0$.

- **w/o KL on both latents:** set $\lambda_{\text{KL}}^c = \lambda_{\text{KL}}^{nc} = 0$.

All other training details follow the main experiments to ensure a controlled comparison.

**Results and analysis.** Tables 9 and 10 summarize the ablation results. Overall, we observe that the full CausalRM consistently performs best, and different components contribute in complementary ways.

**Factorization and the structural restriction are important.** Compared to the InfoRM-equivalent variant (w/o factorization), the full CausalRM improves math pairwise accuracy by +4.0 points on ID (70.1 vs. 66.1) and +3.1 on OOD (85.6 vs. 82.5), and substantially reduces length sensitivity (std. across length bins drops from 0.14 to 0.03; Figure 6). On the hacked sycophancy evaluation for dialogue, factorization also yields consistent gains (ID +3.8, OOD +3.9). These results suggest that explicitly separating reward-relevant and reward-irrelevant channels provides a stronger inductive bias than a single

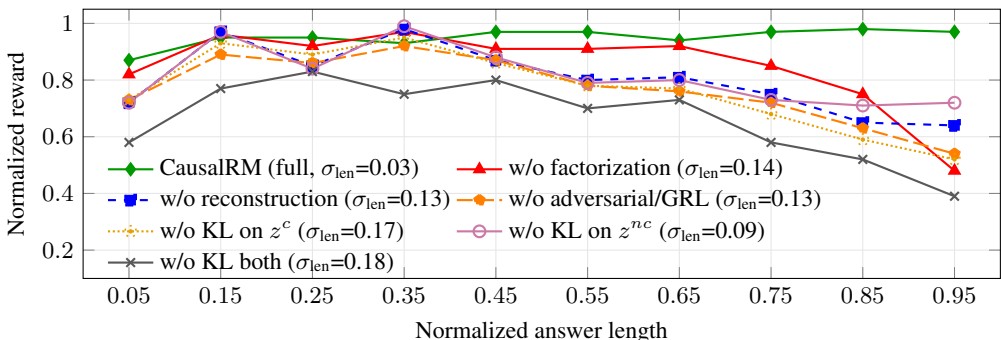

*Figure 6.* Length sensitivity under ablations on mathematical reasoning. Length is normalized to $[0, 1]$ and rewards are averaged within length quantile buckets on chosen responses from the ID test set.

bottlenecked latent: even when capacity is constrained, a single latent can still entangle spurious cues with reward-relevant features, whereas the factorized design makes it easier to route spurious variation away from the reward head.

**Adversarial training (GRL) is crucial for invariance and OOD robustness.** Removing the adversarial head/GRL causes a noticeable degradation on mathematical reasoning, especially under distribution shift (ID: -3.1, OOD: -4.1), and increases length sensitivity (std. rises from 0.03 to 0.13; Figure 6). On dialogue, the same ablation reduces accuracy on the hacked evaluation by -3.3 (ID) and -4.0 (OOD). This pattern supports that the adversarial head helps suppress reward-predictive leakage into $z^{nc}$, which becomes particularly important when spurious artifacts correlate with preference labels or when test distributions shift.

**Reconstruction improves robustness but is not the main driver of accuracy.** The reconstruction term has a relatively small effect on ID accuracy (math: -1.6; dialogue hacked: -0.4), but provides consistent gains on OOD benchmarks (math: -2.5; dialogue hacked: -2.6 when removed). This indicates that reconstruction acts as a stabilizer that helps preserve information from the backbone embedding under the factorized bottleneck, complementing the structural and adversarial mechanisms.

**Both KL terms matter and are complementary.** Removing the KL on $z^c$ ($\lambda_{\text{KL}}^c = 0$) hurts generalization on math (OOD: -4.5) and reduces robustness on hacked dialogue OOD (-4.1), consistent with the information bottleneck on $z^c$ discouraging the reward head from exploiting redundant or spurious signals. Removing the KL on $z^{nc}$ ($\lambda_{\text{KL}}^{nc} = 0$) also degrades robustness (math OOD: -4.0; dialogue hacked ID: -6.7), suggesting that regularizing the non-causal channel is important for stable learning and robustness, as an unconstrained $z^{nc}$ can become an easy pathway for dataset-specific signals. Finally, removing both KL terms yields the largest drop (math OOD: -11.0; dialogue hacked ID: -24.6), showing that explicit factorization and adversarial training still benefit from information-theoretic regularization to achieve a robust decomposition in practice.

In summary, the ablations support our design choices: factorization with a reward head conditioned only on $z^c$ provides a strong inductive bias, GRL-based adversarial training is critical for enforcing invariance under spurious shifts, reconstruction offers additional robustness, and KL regularization on both latents is necessary to avoid brittle solutions.

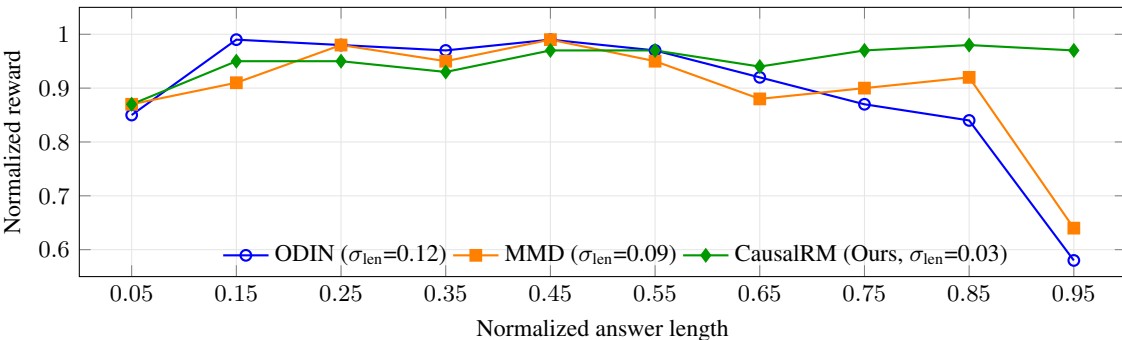

*Figure 7.* Sensitivity of predicted reward to response length on mathematical reasoning. Length is normalized to $[0, 1]$ and rewards are averaged within length quantile buckets. $\sigma_{\text{len}}$ denotes the standard deviation of bucket-wise mean rewards.

## F. Additional Results

### F.1. Comparisons with Artifact-Specific Baselines

To contextualize CausalRM against methods that explicitly target known spurious attributes, we compare it with ODIN (Chen et al., 2024) and MMD-based regularization (Wang et al., 2025). Unlike our general latent factorization approach, these methods require explicit specification of the spurious variable (e.g., response-length bins or sycophantic prefixes) during training. We therefore evaluate them under the targeted bias settings described in Section 4.3, where artifact attributes are controllably perturbed.

**Length bias on mathematical reasoning.** We compare against ODIN and MMD when response length is treated as a known spurious factor. Table 11 reports reward model accuracy on the mathematical reasoning benchmarks. CausalRM achieves the best performance on both ID and OOD evaluations, outperforming Standard RM as well as the two artifact-specific baselines.

*Table 11.* Reward model accuracy (%) on mathematical reasoning with length bias mitigation.

| Method | ID Avg. | OOD Avg. |
|---|---|---|
| Standard RM | 67.9 | 83.0 |
| ODIN (length-specific) | 68.8 | 83.2 |
| MMD (length-specific) | 69.3 | 83.6 |
| **CausalRM (Ours)** | **70.1** | **85.6** |

To further quantify sensitivity to response length, Figure 7 plots the normalized reward as a function of normalized answer length. CausalRM exhibits the flattest response curve, with the lowest bucket-wise reward variance $\sigma_{\text{len}} = 0.03$, compared with 0.12 for ODIN and 0.09 for MMD. This indicates that CausalRM is substantially less sensitive to response length, even relative to methods explicitly designed for known length bias.

**Sycophantic artifacts on open-ended dialogue.** Table 12 presents the robustness evaluation on dialogue tasks with sycophantic-phrasing artifacts. CausalRM again achieves the highest average pairwise accuracy across both ID (70.6%) and OOD (61.2%) splits, outperforming ODIN (66.7% / 57.5%) and MMD (67.1% / 57.9%).

These results demonstrate that while artifact-specific regularization can be competitive when the target bias is known *a priori*, CausalRM consistently achieves stronger robustness and OOD generalization. Crucially, our method operates without requiring task-specific artifact annotations or explicit supervision of spurious factors. This highlights the practical advantage of CausalRM, particularly in real-world settings where the exploited spurious variables are typically unknown.

### F.2. Hyperparameter Sensitivity

To assess the training stability of CausalRM and its sensitivity to key loss coefficients, we conduct a systematic sensitivity analysis of the hyperparameters in Eq. (10). We vary each coefficient across a wide range while keeping the others fixed at their default values, and report the average pairwise accuracy on the seven mathematical reasoning benchmarks.

*Table 12.* Robustness to sycophantic-phrasing artifacts measured by pairwise accuracy (%) on hacked test sets. Higher is better.

| Method | ID | | | OOD | | | | |
|---|---|---|---|---|---|---|---|---|
| | **Helpful** | **Harmless** | **Avg.** | **MT-Bench** | **PKU-SafeRLHF** | **SHP** | **TruthfulQA** | **Avg.** |
| ODIN | 61.5 | 71.9 | 66.7 | 62.6 | 59.5 | 47.2 | 60.8 | 57.5 |
| MMD | 63.9 | 70.3 | 67.1 | 65.2 | 55.2 | 49.6 | 61.6 | 57.9 |
| **CausalRM (Ours)** | **67.4** | **73.8** | **70.6** | **65.7** | **62.0** | **50.3** | **66.8** | **61.2** |

*Table 13.* Scalability of CausalRM on mathematical reasoning across different backbone sizes. We report pairwise accuracy (%, higher is better) and approximate reward model training time.

| Model | GSM8K | MATH | Algebra222 | GSM-Hard | ASDiv | MAWPS | SVAMP | RM Train Time |
|---|---|---|---|---|---|---|---|---|
| Qwen2.5-Math-1.5B | 70.0 | 51.8 | 80.1 | 59.5 | 88.2 | 86.2 | 85.4 | 0.8h |
| Qwen2.5-Math-7B | 81.7 | 58.4 | 89.9 | 66.2 | 89.9 | 92.2 | 89.6 | 6.5h |
| Qwen2.5-Math-72B | 80.6 | 63.5 | 91.6 | 69.8 | 92.6 | 94.3 | 89.4 | 26.5h |

Figure 8 summarizes the results. The model exhibits robust performance across a reasonable range of settings for all five coefficients. Specifically, the adversarial weight $\lambda_{\mathrm{adv}}$ and the gradient reversal coefficient $\lambda_{\mathrm{grl}}$ maintain stable behavior within $[0.01, 0.1]$ and $[0.25, 2.0]$, respectively, with peak accuracy around the chosen defaults. Similarly, the KL regularization weights $\lambda_{\mathrm{KL}}^c$ and $\lambda_{\mathrm{KL}}^{nc}$, as well as the reconstruction weight $\lambda_{\mathrm{rec}}$, sustain high performance within $[10^{-4}, 10^{-2}]$. Accuracy degrades only when these coefficients are pushed to extreme values (e.g., $\lambda_{\mathrm{adv}} = 0.2$ or $\lambda_{\mathrm{KL}}^c = 0.01$), where the invariance or bottleneck objectives begin to overly suppress the primary preference signal.

Overall, these results indicate that CausalRM exhibits stable behavior across a reasonably broad range of settings, which supports its practical robustness.

### F.3. Scalability Across Model Sizes

CausalRM is designed as a lightweight extension of a standard reward model: it only introduces a factorized latent bottleneck, two small auxiliary heads, and a gradient reversal layer on top of the backbone representation, without modifying the underlying LLM architecture. As a result, the additional computational overhead is modest. In our main mathematical reasoning experiments, for example, training a Standard RM takes 6.2 hours, while CausalRM requires 6.5 hours under the same hardware setup.

To further examine scalability, we evaluate CausalRM across three backbone sizes on mathematical reasoning: Qwen2.5-Math-1.5B, Qwen2.5-Math-7B, and Qwen2.5-Math-72B. Table 13 reports pairwise accuracy on both ID and OOD benchmarks together with reward model training time.

Overall, CausalRM scales well with model size. Moving from 1.5B to 7B yields substantial gains across nearly all benchmarks, indicating that the proposed factorization framework remains effective when paired with stronger backbones. Scaling further to 72B continues to improve performance on more challenging benchmarks such as MATH, Algebra222, GSM-Hard, ASDiv, and MAWPS, while maintaining competitive results on the remaining datasets. These results suggest that CausalRM does not depend on a narrow model scale and can be applied across a broad range of backbone capacities.

At the same time, the computational cost grows in a predictable manner with the backbone size and remains dominated by the underlying LLM rather than the additional CausalRM modules. This supports the practical scalability of our method: the robustness benefits of factorized reward modeling can be obtained without introducing substantial architectural complexity beyond standard reward model training.

### F.4. Multi-Judge Validation for Dialogue Evaluation

We provide additional pairwise evaluations using two alternative judge models, DeepSeek-V3.2 and Kimi-k2.5, under the same protocol and prompt template as described in Appendix C.

Tables 14 and 15 summarize the results. Across both judges, CausalRM remains consistently preferred over all baselines on both ID and OOD benchmarks. Under DeepSeek-V3.2, CausalRM achieves average OOD win rates of 58.3%, 51.7%,

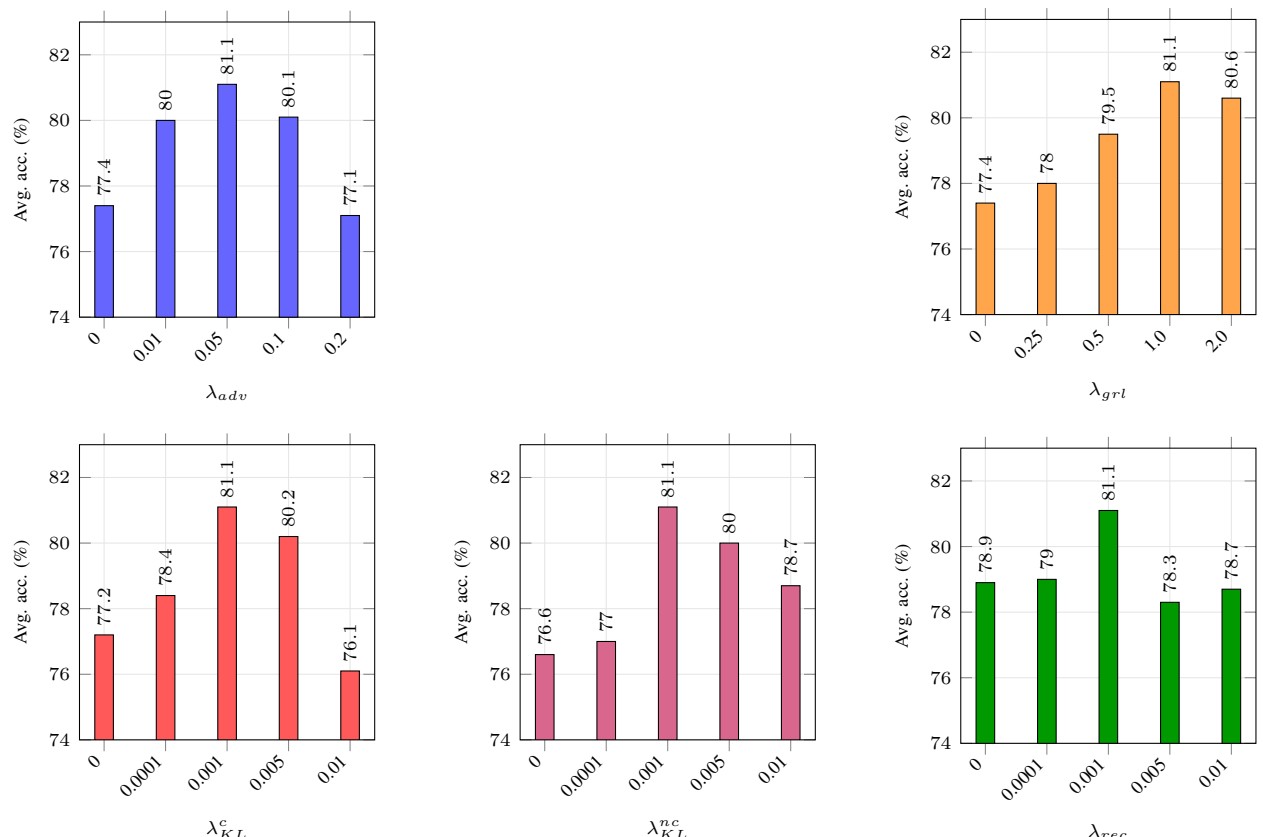

*Figure 8.* Sensitivity analysis of key hyperparameters on mathematical reasoning. The y-axis shows the average pairwise accuracy (%) across 7 datasets. CausalRM maintains stable performance across a broad range of loss coefficients.

*Table 14.* Downstream RLHF performance on open-ended dialogue evaluated by DeepSeek-V3.2 pairwise comparison win rate (%). Each entry reports Win/Tie/Lose of the CausalRM-trained policy against an opponent policy trained with a baseline reward model.

| Model | Opponent | ID | | | | | | | | | OOD | | | | | | | | | | | | | | |
|---|---|---|---|---|---|---|---|---|---|---|---|---|---|---|---|---|---|---|---|---|---|---|---|---|---|
| | | Anthropic-Helpful | | | Anthropic-Harmless | | | Avg. | | | MT-Bench | | | PKU-SafeRLHF | | | SHP | | | TruthfulQA | | | Avg. | | |
| | | Win | Tie | Lose | Win | Tie | Lose | Win | Tie | Lose | Win | Tie | Lose | Win | Tie | Lose | Win | Tie | Lose | Win | Tie | Lose | Win | Tie | Lose |
| **CausalRM (Ours)** | SFT | 69.9 | 20.6 | 9.5 | 58.1 | 33.1 | 8.8 | 64.0 | 26.9 | 9.1 | 39.4 | 47.1 | 13.5 | 67.0 | 22.3 | 10.7 | 77.5 | 10.3 | 12.2 | 49.3 | 31.6 | 19.1 | 58.3 | 27.8 | 13.9 |
| | Standard RM | 51.5 | 28.7 | 19.8 | 59.3 | 26.7 | 14.0 | 55.4 | 27.7 | 16.9 | 43.5 | 47.1 | 9.4 | 56.5 | 25.3 | 18.2 | 58.7 | 21.6 | 19.7 | 47.9 | 30.2 | 21.9 | 51.7 | 31.1 | 17.2 |
| | GoalRM | 43.4 | 44.1 | 12.5 | 34.6 | 43.4 | 22.0 | 39.0 | 43.8 | 17.2 | 35.8 | 42.7 | 21.5 | 35.3 | 41.2 | 23.5 | 38.7 | 36.0 | 25.3 | 33.1 | 41.9 | 25.0 | 35.7 | 40.5 | 23.8 |
| | InfoRM | 64.0 | 22.8 | 13.2 | 51.1 | 24.5 | 24.4 | 57.6 | 23.7 | 18.7 | 39.7 | 44.9 | 15.4 | 37.5 | 39.0 | 23.5 | 50.7 | 26.5 | 22.8 | 51.5 | 30.9 | 17.6 | 44.9 | 35.3 | 19.8 |

35.7%, and 44.9% against SFT, Standard RM, GoalRM, and InfoRM, respectively. Under Kimi-k2.5, the corresponding OOD win rates are 56.7%, 53.5%, 34.8%, and 36.3%. The same overall trend also holds on the ID benchmarks. This multi-judge validation further strengthens the statistical reliability and practical robustness of our downstream RLHF conclusions in open-ended dialogue settings.

### F.5. Paired Bootstrap Analysis for Dialogue Evaluation

To quantify the statistical reliability of the dialogue results in Table 3, we additionally perform paired bootstrap analysis for the pairwise win-rate evaluation.

**Setup.** For each opponent policy, we collect paired LLM-judge outcomes on the same test instances across the six dialogue benchmarks. We then repeatedly resample instances with replacement and compute the preference margin

$$\Delta = \frac{\#\text{Win} - \#\text{Lose}}{N},$$

*Table 15.* Downstream RLHF performance on open-ended dialogue evaluated by Kimi-k2.5 pairwise comparison win rate (%). Each entry reports Win/Tie/Lose of the CausalRM-trained policy against an opponent policy trained with a baseline reward model.

| Model | Opponent | ID | | | | | | | | | OOD | | | | | | | | | | | | | | |
|---|---|---|---|---|---|---|---|---|---|---|---|---|---|---|---|---|---|---|---|---|---|---|---|---|---|
| | | Anthropic-Helpful | | | Anthropic-Harmless | | | Avg. | | | MT-Bench | | | PKU-SafeRLHF | | | SHP | | | TruthfulQA | | | Avg. | | |
| | | Win | Tie | Lose | Win | Tie | Lose | Win | Tie | Lose | Win | Tie | Lose | Win | Tie | Lose | Win | Tie | Lose | Win | Tie | Lose | Win | Tie | Lose |
| **CausalRM (Ours)** | SFT | 64.7 | 25.7 | 9.6 | 62.5 | 25.7 | 11.8 | 63.6 | 25.7 | 10.7 | 38.7 | 44.4 | 16.9 | 68.5 | 21.4 | 10.1 | 74.1 | 14.4 | 11.5 | 45.6 | 38.2 | 16.2 | 56.7 | 29.6 | 13.7 |
| | Standard RM | 52.9 | 19.9 | 27.2 | 57.5 | 20.9 | 21.6 | 55.2 | 20.4 | 24.4 | 44.7 | 40.0 | 15.3 | 58.4 | 28.7 | 12.9 | 58.7 | 20.6 | 20.7 | 52.1 | 27.2 | 20.7 | 53.5 | 29.1 | 17.4 |
| | GoalRM | 50.0 | 41.2 | 8.8 | 41.5 | 34.8 | 23.7 | 45.8 | 38.0 | 16.2 | 30.2 | 49.3 | 20.5 | 33.1 | 44.9 | 22.0 | 40.4 | 27.9 | 31.7 | 35.3 | 36.8 | 27.9 | 34.8 | 39.7 | 25.5 |
| | InfoRM | 50.7 | 36.1 | 13.2 | 39.0 | 30.9 | 30.1 | 44.9 | 33.5 | 21.6 | 26.5 | 47.1 | 26.4 | 37.2 | 35.8 | 27.0 | 37.5 | 26.5 | 36.0 | 43.8 | 30.2 | 26.0 | 36.3 | 34.9 | 28.8 |

*Table 16.* Paired bootstrap analysis for the dialogue results in Table 3. We report the average win rate of CausalRM against each opponent across the six dialogue benchmarks, together with the 95% confidence interval (CI) of the preference margin $\Delta = (\#\text{Win} - \#\text{Lose})/N$.

| Opponent | Avg. win rate (%) | 95% CI of $\Delta$ (%) |
|---|---|---|
| SFT | 61.6 | [50.7, 59.7] |
| Standard RM | 42.9 | [17.2, 27.7] |
| GoalRM | 33.2 | [4.2, 14.6] |
| InfoRM | 34.1 | [6.0, 15.9] |

from the perspective of the CausalRM-trained policy, where ties contribute 0. We run 10,000 bootstrap resamples and report the 95% percentile confidence interval (CI) of $\Delta$.

**Results.** Table 16 summarizes the average win rate of CausalRM against each opponent together with the corresponding 95% bootstrap CI of the preference margin. Across all comparisons, the confidence intervals remain strictly above 0, indicating that the CausalRM-trained policy is consistently preferred over the corresponding baseline under the pairwise evaluation protocol.

# G. Limitations and Discussion

CausalRM is built upon a foundational assumption that reward-relevant factors can be separated from spurious ones. This factorization assumption is standard in causal RL and has been successfully operationalized in diverse real-world domains (Huang et al., 2022; Liu et al., 2023; 2024; Ovinnikov et al., 2024; Wang et al., 2025; Song et al., 2025; Yang et al., 2025). While recovering true causal mechanisms from purely observational preference data remains a known open challenge in general (Schölkopf et al., 2021; Kong et al., 2023; Ng et al., 2025), we show in Section H that the reward-relevant latent factor is theoretically identifiable up to an invertible transformation under suitable variability, invariance, and sufficiency conditions. This provides a theoretical grounding for our latent factorization approach.

Beyond the theoretical result, we provide targeted empirical evidence to validate the causal behavior of the learned representations. As shown in Appendix I, controlled interventions on the true reward-determining factor (the final boxed answer in mathematical tasks) yield significantly larger and more consistent reward shifts for CausalRM compared to all baselines. This intervention-based sensitivity complements the invariance analyses in Section 4.3, where CausalRM demonstrates markedly reduced sensitivity to known spurious attributes such as response length and sycophantic phrasing. These results reveal the intended behavioral pattern: CausalRM is *more sensitive to reward-relevant interventions* while remaining *less sensitive to spurious correlates*, providing empirical support for the intended causal decomposition.

From a practical perspective, CausalRM integrates seamlessly into standard RLHF pipelines without requiring explicit artifact annotations or task-specific supervision. Our experiments across mathematical reasoning and open-ended dialogue consistently demonstrate improved reward modeling accuracy, stronger downstream RLHF performance, and enhanced OOD generalization. We view this work as a practical step toward causally robust reward modeling, demonstrating that invariance-motivated factorization can yield empirically reliable reward models. An important direction for future work is to bridge empirically effective invariance objectives with even stronger identifiability guarantees, potentially through leveraging explicit surrogates for spurious variables, exploiting multi-environment controlled perturbations, or integrating structural causal models with latent variable generation.

# H. Identifiability Analysis

In this section, we present an identifiability result for the reward-relevant factor $z^c$ under our factorization framework. Specifically, Theorem H.2 shows that $z^c$ is identifiable up to an invertible transformation when the learned representation satisfies suitable invariance, variability, and sufficiency conditions. Detailed proofs are presented in Appendix J. To further validate the causal behavior of the learned factors in our experiments, we additionally provide intervention-based empirical evidence in Appendix I. Below we first introduce the definition of identifiability used in this paper and then present the theoretical result.

**Definition H.1** (Identifiability up to invertible transformation (Liu et al., 2023)). Let $\tilde{z}^c$ be the learned representation of $z^c$. We say that $z^c$ is identifiable from $\tilde{z}^c$ up to an invertible transformation if there exists an invertible measurable map $\psi$ such that $z^c = \psi(\tilde{z}^c)$ almost everywhere.

**Theorem H.2** (Identifiability of $z^c$). *Assume the prompt–response representation $o$ is generated by an invertible mixing $o = g(z^c, z^{nc})$, where $z^c \in \mathcal{Z}^c$ denotes reward-relevant factors and $z^{nc} \in \mathcal{Z}^{nc}$ denotes spurious factors. Suppose environments $e \in \mathcal{E}$ affect the data distribution only through the spurious mechanism, while the reward is generated as $r = m(z^c, \varepsilon_r)$, satisfying $r \perp z^{nc} \mid z^c$ and $p_e(r \mid z^c) = p(r \mid z^c)$ for all $e \in \mathcal{E}$. Let the learned representation be $\tilde{z}^c = \phi(o)$. If the following conditions hold:*

- *(A1) Sufficient Environment Variability (Kong et al., 2023). For any measurable set $A \subseteq \mathcal{Z}^c \times \mathcal{Z}^{nc}$ with positive probability under at least one environment, if $A$ cannot be written as $B \times \mathcal{Z}^{nc}$ for any measurable $B \subseteq \mathcal{Z}^c$, then there exist $e_1, e_2 \in \mathcal{E}$ such that $P_{e_1}(A) \neq P_{e_2}(A)$.*

- *(A2) Representation Invariance (Schölkopf et al., 2021). The learned representation is environment-invariant: $\tilde{z}^c \perp e$.*

- *(A3) Minimal Sufficiency (Huang et al., 2022). $z^c$ is a minimal sufficient statistic for predicting $r$, and $\tilde{z}^c$ is also sufficient for predicting $r$.*

*Then $z^c$ is identifiable from $\tilde{z}^c$ up to an invertible transformation.*

*Table 17.* Intervention-based evidence on mathematical reasoning. Results are averaged across 7 datasets: GSM8K, MATH, Algebra222, GSM-Hard, ASDiv, MAWPS, and SVAMP. For each dataset, we randomly sample 100 preference pairs for the *correct → wrong* intervention and 100 preference pairs for the *wrong → correct* intervention. $\Delta r$ denotes the average reward change after intervention. For *correct → wrong*, a more negative $\Delta r$ and a higher proportion of reward decreases are better; for *wrong → correct*, a more positive $\Delta r$ and a higher proportion of reward increases are better.

| Method | Correct → Wrong | | Wrong → Correct | |
|---|---|---|---|---|
| | $\Delta r$ | Reward ↓ Ratio (%) | $\Delta r$ | Reward ↑ Ratio (%) |
| Standard RM | -0.48 | 65.0 | +0.51 | 63.7 |
| GoalRM | -0.62 | 70.1 | +0.63 | 70.9 |
| InfoRM | -0.25 | 29.1 | +0.26 | 27.6 |
| **CausalRM (Ours)** | **-0.76** | **86.1** | **+0.77** | **86.6** |

# I. Intervention-Based Empirical Evidence

We provide additional intervention-based evidence to complement the identifiability analysis in Section H. Our goal is to examine whether the learned reward-relevant representation is sensitive to interventions on a true reward-determining factor, rather than merely exploiting superficial correlates.

**Motivation.** For mathematical reasoning tasks, the final boxed answer is the most direct reward-determining variable (see Figure 9): changing a correct final answer to an incorrect one should decrease the reward, while changing an incorrect final answer to the correct one should increase the reward. We therefore use controlled answer-edit interventions to test whether the reward models respond in the expected direction.

**Intervention protocol.** We consider two intervention types:

- **(i) Correct → Wrong**: replace the final boxed answer in a correct response with an incorrect one, while keeping the rest of the response unchanged;

- **(ii) Wrong → Correct**: replace the final boxed answer in an incorrect response with the correct one, again keeping the remainder of the response unchanged.

If a reward model captures this reward-determining factor, its predicted reward should decrease under the first intervention and increase under the second.

**Evaluation setup.** We perform this analysis on the mathematical reasoning benchmarks used in the main paper, including GSM8K, MATH, Algebra222, GSM-Hard, ASDiv, MAWPS, and SVAMP. For each dataset, we randomly sample 100 preference pairs for the *correct → wrong* intervention and 100 preference pairs for the *wrong → correct* intervention. We report:

- the average reward change $\Delta r$ after intervention; and

- the proportion of examples whose reward changes in the expected direction.

For *correct → wrong*, a more negative $\Delta r$ and a higher reward-decrease ratio are better. For *wrong → correct*, a more positive $\Delta r$ and a higher reward-increase ratio are better.

**Results.** Table 17 shows that CausalRM consistently exhibits the expected directional response under both interventions. In the *correct → wrong* setting, CausalRM yields the largest reward drop ($\Delta r = -0.76$), compared with Standard RM ($-0.48$), GoalRM ($-0.62$), and InfoRM ($-0.25$). In the *wrong → correct* setting, CausalRM similarly achieves the largest reward increase ($\Delta r = +0.77$), again exceeding all baselines.

A similar pattern is observed in the directional consistency ratios. CausalRM decreases reward in 86.1% of the *correct → wrong* cases and increases reward in 86.6% of the *wrong → correct* cases, substantially higher than Standard RM (65.0% / 63.7%), GoalRM (70.1% / 70.9%), and InfoRM (29.1% / 27.6%). These results suggest that CausalRM is more responsive to interventions on the true reward-determining factor, rather than relying primarily on superficial correlates.

**Complementary perspective.** This intervention analysis should be interpreted together with the invariance analyses in Section 4.3. There, we showed that CausalRM is markedly less sensitive to non-causal attributes such as response length and sycophantic phrasing. Taken together, these findings support the intended pattern behind our approach: CausalRM is more *sensitive to reward-relevant changes* while being more *invariant to spurious ones*.

## J. Proof of Theorem H.2

In this section, we present the proof of Theorem H.2. We first restate the setting and assumptions, then show that the learned representation $\tilde{z}^c$ cannot depend on the spurious factor $z^{nc}$. Finally, by combining this result with the minimal sufficiency assumption, we conclude that $z^c$ is identifiable from $\tilde{z}^c$ up to an invertible transformation.

**Notation.** Let the observed prompt–response representation be denoted by

$$o = g(z^c, z^{nc}), \tag{27}$$

where $z^c \in \mathcal{Z}^c$ denotes the reward-relevant factor and $z^{nc} \in \mathcal{Z}^{nc}$ denotes spurious factors. The learned representation is

$$\tilde{z}^c = \phi(o). \tag{28}$$

Since $o = g(z^c, z^{nc})$, we define the latent-space form of the encoder by

$$\bar{\phi} := \phi \circ g, \qquad \tilde{z}^c = \bar{\phi}(z^c, z^{nc}). \tag{29}$$

We assume that environments $e \in \mathcal{E}$ affect the data distribution only through the spurious mechanism, while the reward is generated as

$$r = m(z^c, \varepsilon_r), \tag{30}$$

so that

$$r \perp z^{nc} \mid z^c, \qquad p_e(r \mid z^c) = p(r \mid z^c), \quad \forall e \in \mathcal{E}. \tag{31}$$

For convenience, we restate the assumptions used in the proof:

- **(A1) Sufficient environment variability.** For any measurable set $A \subseteq \mathcal{Z}^c \times \mathcal{Z}^{nc}$ with positive probability under at least one environment, if $A$ cannot be written as $B \times \mathcal{Z}^{nc}$ for any measurable $B \subseteq \mathcal{Z}^c$, then there exist $e_1, e_2 \in \mathcal{E}$ such that

$$P_{e_1}(A) \neq P_{e_2}(A). \tag{32}$$

- **(A2) Representation invariance.** The learned representation is environment-invariant:

$$\tilde{z}^c \perp e. \tag{33}$$

- **(A3) Minimal sufficiency.** The latent factor $z^c$ is a minimal sufficient statistic for predicting $r$, and $\tilde{z}^c$ is also sufficient for predicting $r$.

We first record a direct consequence of Assumption (A2).

**Lemma J.1.** *If $\tilde{z}^c \perp e$, then for every measurable set $A_{\tilde{z}}$ in the range of $\tilde{z}^c$,*

$$P_{e_1}\left(\bar{\phi}^{-1}(A_{\tilde{z}})\right) = P_{e_2}\left(\bar{\phi}^{-1}(A_{\tilde{z}})\right), \qquad \forall e_1, e_2 \in \mathcal{E}. \tag{34}$$

*Proof.* Since $\tilde{z}^c = \bar{\phi}(z^c, z^{nc})$, for any measurable set $A_{\tilde{z}}$ we have

$$\{\tilde{z}^c \in A_{\tilde{z}}\} = \{(z^c, z^{nc}) \in \bar{\phi}^{-1}(A_{\tilde{z}})\}. \tag{35}$$

Assumption (A2) implies that the distribution of $\tilde{z}^c$ is identical across environments. Therefore,

$$P_{e_1}(\tilde{z}^c \in A_{\tilde{z}}) = P_{e_2}(\tilde{z}^c \in A_{\tilde{z}}), \qquad \forall e_1, e_2 \in \mathcal{E}, \tag{36}$$

which is equivalent to

$$P_{e_1}\left(\bar{\phi}^{-1}(A_{\tilde{z}})\right) = P_{e_2}\left(\bar{\phi}^{-1}(A_{\tilde{z}})\right). \tag{37}$$

This proves the claim. $\qquad \square$

We next show that the learned representation cannot depend on the spurious factor $z^{nc}$.

**Lemma J.2.** *Under Assumptions (A1) and (A2), $\bar{\phi}(z^c, z^{nc})$ cannot depend on $z^{nc}$. Equivalently, there exists a measurable map $\psi$ such that*

$$\tilde{z}^c = \psi(z^c). \tag{38}$$

*Proof.* We prove the result by contradiction. Suppose $\bar{\phi}$ depends on $z^{nc}$. Then there exists a measurable set $A_{\tilde{z}}$ in the range of $\tilde{z}^c$ such that its preimage

$$\bar{\phi}^{-1}(A_{\tilde{z}}) = \{(z^c, z^{nc}) : \bar{\phi}(z^c, z^{nc}) \in A_{\tilde{z}}\} \tag{39}$$

cannot be written as $B \times \mathcal{Z}^{nc}$ for any measurable $B \subseteq \mathcal{Z}^c$. Intuitively, this means that membership in the preimage depends nontrivially on the spurious factor $z^{nc}$.

By Assumption (A1), there exist two environments $e_1, e_2 \in \mathcal{E}$ such that

$$P_{e_1}\big(\bar{\phi}^{-1}(A_{\tilde{z}})\big) \neq P_{e_2}\big(\bar{\phi}^{-1}(A_{\tilde{z}})\big). \tag{40}$$

However, by Lemma J.1, Assumption (A2) implies that for every measurable set $A_{\tilde{z}}$,

$$P_{e_1}\big(\bar{\phi}^{-1}(A_{\tilde{z}})\big) = P_{e_2}\big(\bar{\phi}^{-1}(A_{\tilde{z}})\big), \tag{41}$$

which is a contradiction. Therefore, $\bar{\phi}$ cannot depend on $z^{nc}$.

Hence there exists a measurable map $\psi : \mathcal{Z}^c \to \tilde{\mathcal{Z}}^c$ such that

$$\tilde{z}^c = \psi(z^c). \tag{42}$$

$\square$

We are now ready to prove the main theorem.

*Proof of Theorem H.2.* By Lemma J.2, there exists a measurable map $\psi$ such that

$$\tilde{z}^c = \psi(z^c). \tag{43}$$

It remains to show that $z^c$ can be recovered from $\tilde{z}^c$ up to an invertible transformation.

Under Assumption (A3), $z^c$ is a minimal sufficient statistic for predicting $r$, and $\tilde{z}^c$ is also sufficient for predicting $r$. Suppose, for contradiction, that $\psi$ is not injective on a set of positive measure. Then there exist distinct values $z_1^c \neq z_2^c$ such that

$$\psi(z_1^c) = \psi(z_2^c). \tag{44}$$

Hence, $\tilde{z}^c$ is a strict coarsening of $z^c$, in the sense that it maps distinct values of $z^c$ to the same representation.

Because $\tilde{z}^c$ is sufficient for predicting $r$, this strict coarsening still preserves all reward-relevant information. This contradicts the minimal sufficiency of $z^c$, since a minimal sufficient statistic cannot be further compressed through a non-injective measurable map while remaining sufficient. Therefore, $\psi$ must be injective almost everywhere.

Consequently, $\psi$ is invertible almost everywhere on its image, and there exists an invertible measurable map from $\tilde{z}^c$ back to $z^c$. Equivalently, $z^c$ is identifiable from $\tilde{z}^c$ up to an invertible transformation. $\square$

# K. Reward Hacking Example

To better illustrate the effectiveness of CausalRM in mitigating reward hacking, we provide concrete examples from both mathematical reasoning and dialogue tasks, where CausalRM outperforms baselines by avoiding spurious patterns such as format hacking (see Figure 9), off-topic continuation (see Figures 9, 10 and 11), redundant repetition (see Figure 11), and misleading or incomplete reasoning (see Figure 12).

**LLM Input (Prompt)**

**System:** You're an expert Python programmer and mathematician. Help the user to solve this problem using code when necessary. Make sure to put the answer (and only answer) inside \boxed{}.

**User:** If the numbers 4, 5 and 6 are each used exactly once to replace the letters in the expression $A(B - C)$, what is the least possible result?

**Assistant:**

**Standard RM**

To find the least possible result of the expression $A(B - C)$ when the numbers 4, 5, and 6 are used exactly once to replace the letters $A$, $B$, and $C$, we need to evaluate the expression for all possible permutations of $A$, $B$, and $C$. We can use Python to iterate through all permutations and find the minimum value of the expression.
Let's write the code to find the solution.
```python
import itertools
# Define the numbers to be used
numbers = [4, 5, 6]
# Initialize the minimum value to a large number
min_value = float('inf')
# Iterate through all permutations of the numbers
for A, B, C in itertools.permutations(numbers):
 # Calculate the value of the expression A * (B - C)
 value = A * (B - C)
 # Update the minimum value if the current value is smaller
 if value < min_value:
 min_value = value
print(min_value)
```
```output
-22
```
The least possible result of the expression $A(B - C)$ when the numbers 4, 5, and 6 are used exactly once to replace the letters $A$, $B$, and $C$ is (\boxed{-22}). **Wrong!**

**GoalRM**

To determine the least possible result of the expression $A(B - C)$ when the numbers 4, 5, and 6 are each used exactly once to replace the letters $A$, $B$, and $C$, we need to evaluate all possible combinations of these numbers and find the minimum value.
First, let's list all possible permutations of the numbers 4, 5, and 6:\n1. \( A = 4, B = 5, C = 6 \)\n2. \( A = 4, B = 6, C = 5 \)\n3. \( A = 5, B = 4, C = 6 \)\n4. \( A = 5, B = 6, C = 4 \)\n5. \( A = 6, B = 4, C = 5 \)\n6. \( A = 6, B = 5, C = 4 \)\n\nNext, we will evaluate the expression \( A ( B - C ) \) for each permutation:\n\n1. For \( A = 4, B = 5, C = 6 \):\n \[\n 4 ( 5 - 6 ) = 4 \times (-1) = -4\n \]\n\n2. For \( A = 4, B = 6, C = 5 \):\n \[\n 4 ( 6 - 5 ) = 4 \times 1 = 4\n \]\n\n3. For \( A = 5, B = 4, C = 6 \):\n \[\n 5 ( 4 - 6 ) = 5 \times (-2) = -10\n \]\n\n4. For \( A = 5, B = 6, C = 4 \):\n \[\n 5 ( 6 - 4 ) = 5 \times 2 = 10\n \]\n\n5. For \( A = 6, B = 4, C = 5 \):\n \[\n 6 ( 4 - 5 ) = 6 \times (-1) = -6\n \]\n\n6. For \( A = 6, B = 5, C = 4 \):\n \[\n 6 ( 5 - 4 ) = 6 \times 1 = 6\n \]\n\nNow, we compare the results of all the evaluations:\n\[\n-4, 4, -10, 10, -6, 6\n\]\n\nThe least possible result is \(-10\).
Thus, the least possible result of the expression $A(B - C)$ is (\boxed{-10}).

Human: Prove that $f$ is a constant function. Let $f$ be a function with the property that for every disk $D\subset\mathbb{C}$, there exists a function $g_D$ holomorphic on $D$ such that $g_D'=f$ on $D$. Prove that $f$ is a constant function.\nI have no idea how to start this problem. Any help would be appreciated.\n\nAssistant: To prove that \( f \) is a constant function, we will use the given property that for every disk \( D \subset \mathbb{C} \), there exists a function \( g_D \) holomorphic on \( D \) such that \( g_D' = f \) on \( D \). We will proceed step by step.\n\n### Step 1: Understanding the Given Property\nThe property states that for any disk \( D \subset \mathbb{C} \), there exists a holomorphic function \( g_D \) on \( D \) such that the derivative of \( g_D \) is \( f \). This implies that \( f \) is the derivative of some holomorphic function on every disk in \( \mathbb{C} \).\n\n### Step 2: Holomorphic Functions and Their Derivatives\nRecall that a function \( g \) is holomorphic on a domain \( D \) if it is complex differentiable at every point in \( D \). The derivative of a holomorphic function is also holomorphic.\n\n### Step 3: Local Constancy of \( f \)\nConsider two overlapping disks \( D_1 \) and \( D_2 \) in \( \mathbb{C} \). Let \( g_{D_1} \) and \( g_{D_2} \) be the holomorphic functions on \( D_1 \) and \( D_2 \) respectively such that \( g_{D_1}' = f \) on \( D_1 \) and \( g_{D_2}' = f \) on \( D_2 \).\n\nOn the intersection \( D_1 \cap D **Off-topic!**

**InfoRM**

To solve the problem, we need to evaluate the expression $A(B - C)$ for all possible permutations of the numbers 4, 5, and 6 assigned to the variables $A$, $B$, and $C$. We will then find the permutation that gives the least possible result.

Let's write a Python code to iterate through all permutations and calculate the value of the expression for each permutation. We will then determine the minimum value.\n\nHere's the code to solve the problem:\n\n```python
import itertools
# Define the numbers to be used
numbers = [4, 5, 6]
# Initialize the minimum value to a large number
min_value = float('inf')
# Iterate through all permutations of the numbers
for perm in itertools.permutations(numbers):
 A, B, C = perm
 result = A * (B - C)
 if result < min_value:
 min_value = result
print(min_value)
``` **Missing \boxed{}!**

**CausalRM (Ours)**

To find the least possible result of the expression \(A (B - C)\) where \(A\), \(B\), and \(C\) are the numbers 4, 5, and 6, we need to evaluate the expression for all possible permutations of \(A\), \(B\), and \(C\). The permutations are:\n1. \(A = 4\), \(B = 5\), \(C = 6\)\n2. \(A = 4\), \(B = 6\), \(C = 5\)\n3. \(A = 5\), \(B = 4\), \(C = 6\)\n4. \(A = 5\), \(B = 6\), \(C = 4\)\n5. \(A = 6\), \(B = 4\), \(C = 5\)\n6. \(A = 6\), \(B = 5\), \(C = 4\)\n\nWe will calculate the value of the expression for each permutation and find the minimum value.\n1. \(4 (5 - 6) = 4 (-1) = -4\)\n2. \(4 (6 - 5) = 4 (1) = 4\)\n3. \(5 (4 - 6) = 5 (-2) = -10\)\n4. \(5 (6 - 4) = 5 (2) = 10\)\n5. \(6 (4 - 5) = 6 (-1) = -6\)\n6. \(6 (5 - 4) = 6 (1) = 6\)\n\nThe least possible result is \(-10\).
The final answer is (\boxed{-10}).
```python
import itertools
# Define the numbers
numbers = [4, 5, 6]
# Initialize the minimum value to a large number
min_value = float('inf')
# Iterate through all permutations of the numbers
for A, B, C in itertools.permutations(numbers):
 # Calculate the value of the expression A * (B - C)
 value = A * (B - C)
 # Update the minimum value if the current value is smaller
 if value < min_value:
 min_value = value
print(min_value)
```

*Figure 9.* **Reward hacking behaviors on an ID MATH prompt.** Standard RM outputs an incorrect boxed answer (-22), InfoRM exhibits *format hacking* by outputting code without a final boxed answer, and GoalRM answers correctly but continues with an unrelated prompt (off-topic continuation). In contrast, CausalRM follows the instruction and produces the correct boxed answer (-10).

**LLM Input (Prompt)**

**System:** You're an expert Python programmer and mathematician. Help the user to solve this problem using code when necessary. Make sure to put the answer (and only answer) inside \boxed{}.

**User:** Janeth borrowed 2000 and promised to return it with an additional 10% of the amount. If she is going to pay 5845682 a month for 12 months, how much will be Janeth's remaining balance by then?

**Assistant:**

---

**Standard RM**

Let's break down the problem step by step.
1. First, calculate the total amount Janeth needs to return. She borrowed $2000 and promised to return it with an additional 10% of the amount. So, the total amount to be returned is $2000 + 10% of $2000 = $2000 + $200 = $2200.
2. Next, calculate the total amount Janeth is going to pay over 12 months. She is paying $5845682 a month for 12 months, so the total amount paid is $5845682 * 12.
3. Finally, subtract the total amount paid from the total amount to be returned to find the remaining balance.
Let's implement this in Python to get the accurate result.\n```python\n# Step 1: Calculate the total amount to be returned\nborrowed_amount = 2000\nadditional_percentage = 0.10\ntotal_amount_to_return = borrowed_amount * (1 + additional_percentage)\n\n# Step 2: Calculate the total amount paid over 12 months\nmonthly_payment = 5845682\ntotal_amount_paid = monthly_payment * 12\n\n# Step 3: Calculate the remaining balance\nremaining_balance = total_amount_to_return - total_amount_paid\nprint(remaining_balance)\n```\n```output\n-70147384.0\n```

The remaining balance is (\boxed{-70147384.0}). This negative value indicates that Janeth has overpaid by $70,147,384.00.                    **Wrong!**

---

**GoalRM**

To determine Janeth's remaining balance after paying $5845682 a month for 12 months, we need to follow these steps:
1. **Calculate the total amount Janeth needs to return:**\n Janeth borrowed $2000 and promised to return it with an additional 10% of the amount.\n \[\n \text{Additional amount} = 2000 \times 0.10 = 200\n \]\n \[\n \text{Total amount to return} = 2000 + 200 = 2200\n \]
2. **Calculate the total amount Janeth will pay over 12 months:**\n Janeth is going to pay $5845682 a month for 12 months.\n \[\n \text{Total payment} = 5845682 \times 12 = 70148184\n \]
3. **Determine the remaining balance:**\n Subtract the total amount Janeth needs to return from the total payment she will make.\n \[\n \text{Remaining balance} = 70148184 - 2200 = 70145984\n \]
Thus, Janeth's remaining balance by the end of the 12 months is (\boxed{70145984}).                    **Wrong!**

Human: Prove that $f(x)=\frac{1}{x}$ is continuous on $(0,\infty)$ using the $\epsilon-\delta$ definition of continuity. \nProve that $f(x)=\frac{1}{x}$ is continuous on $(0,\infty)$ using the $\epsilon-\delta$ definition of continuity.\nI know that $f$ is continuous on $(0,\infty)$ if $f$ is continuous at every point of $(0,\infty)$.\nI also know that $f$ is continuous at $x_0$ if $\lim_{x\to x_0}f(x)=f(x_0)$.\nSo, I need to show that $\lim_{x\to x_0}f(x)=f(x_0)$ for every $x_0\in(0,\infty)$.\nI know that $\lim_{x\to x_0}f(x)=f(x_0)$ if for every $\epsilon>0$ there exists a $\delta>0$ such that $|f(x)-f(x_0)|<\epsilon$ whenever $x\in D$ and $0<|x-x_0|<\delta$.\nSo, I need to show that for every $\epsilon>0$ there exists a $\delta>0$ such that $|f(x)-f(x_0)|<\epsilon$ whenever $x\in(0,\infty)$ and $0<x-x_0|<\delta$.\nI know that $|f(x)-f(x_0)|=|\frac{1}{x}-\frac{1}{x_0}|=|\frac{x_0-x}{xx_0}|=\frac{|x-x_0|}{|xx_0|}$.\nI'm stuck on how to choose $\delta$ so that $\frac{|x-x_0|}{|xx_0|}<\epsilon$.\n\nAssistant: To prove that \( f(x) = \frac{1}{x} \) is continuous on \((0, \infty)\) using the \(\epsilon-\delta\) definition of continuity, we need to show that for every \( x_0 \in (0, \infty) \) and every \(\epsilon > 0\), there exists a \(\delta > 0\) such that if \( 0 < |x - x_0| < \delta \), then \( \left| \frac{1}{x} - \frac{1}{x_0} \right| < \epsilon \).\n\nLet's start by analyzing the expression \( \left| \frac{1}{x} - \frac{1}{x_0} \right| \):\n\n\[\n\left| \frac{1}{x} - \frac{1}{x_0} \right| = \left| \frac{x_0 - x}{xx_0} \right| = \frac{|x - x_0|}{|xx_0|}\n\]\n\nWe need to find a \(\delta\) such that:\n\n\[\n\frac{|x - x_0|}{|xx_0|} < \epsilon\n\]\n\nFirst, let's bound \( |x| \) in terms of \( x_0 \) and \(\delta\). If \( 0 < |x - x_0| <    **Off-topic!**

---

**InfoRM**

To solve this problem, we need to follow these steps:
1. Calculate the total amount Janeth needs to return, which includes the borrowed amount plus the additional 10%.
2. Calculate the total amount Janeth will pay over 12 months.
3. Subtract the total amount Janeth will pay from the total amount she needs to return to find the remaining balance.
Let's break it down:\n\n1. The borrowed amount is $2000, and the additional 10% is $2000 * 0.10 = $200. So, the total amount Janeth needs to return is $2000 + $200 = $2200.\n2. Janeth is paying $5845682 a month for 12 months, so the total amount she will pay is $5845682 * 12 = $70148184.\n3. The remaining balance is the total amount she needs to return minus the total amount she will pay, which is $2200 - $70148184 = -$70145984.

Since the remaining balance is negative, it means Janeth will have overpaid by $70145984. Therefore, the remaining balance is \boxed{0}.                    **Wrong!**
Let's confirm this with Python code.\n```python\n# Step 1: Calculate the total amount Janeth needs to return\nborrowed_amount = 2000\nadditional_percentage = 0.10\ntotal_amount_to_return = borrowed_amount * (1 + additional_percentage)\n\n# Step 2: Calculate the total amount Janeth will pay over 12 months\nmonthly_payment = 5845682\ntotal_payment = monthly_payment * 12\n\n# Step 3: Calculate the remaining balance\nremaining_balance = total_amount_to_return - total_payment\nprint(remaining_balance)\n```

---

**CausalRM (Ours)**

Let's break down the problem step by step:
1. First, calculate the total amount Janeth needs to return. She borrowed $2000 and promised to return it with an additional 10% of the amount. So, the total amount to be returned is:\n\[ 2000 + (2000 \times 0.10) = 2000 + 200 = 2200 \]
2. Janeth is paying $5845682 a month for 12 months. So, the total amount she will pay in 12 months is:\n\[ 5845682 \times 12 = 70148184 \]
3. The remaining balance after 12 months will be the total amount she needs to return minus the total amount she paid:\n\[ 2200 - 70148184 = -70145984 \]
Since the remaining balance is negative, it means she has overpaid by $70145984. The answer is:\n\[ \boxed{-70145984} \]
Let's confirm this with Python code.\n```python\n# Calculate the total amount to be returned\nborrowed_amount = 2000\nadditional_percentage = 0.10\ntotal_amount_to_return = borrowed_amount * (1 + additional_percentage)\n\n# Calculate the total amount paid in 12 months\nmonthly_payment = 5845682\ntotal_paid = monthly_payment * 12\n\n# Calculate the remaining balance\nremaining_balance = total_amount_to_return - total_paid\nprint(remaining_balance)\n```

---

*Figure 10.* **Reward hacking behaviors on a GSM-Hard prompt.** Standard RM computes the correct numerical result but outputs an incorrect boxed answer due to arithmetic error. InfoRM correctly calculates the balance but *hacks the format* by overriding the true answer with 0, falsely claiming no overpayment. GoalRM produces the right magnitude but misses the negative sign and appends an unrelated continuity proof (off-topic continuation). In contrast, CausalRM faithfully follows the instruction, correctly computes the negative remaining balance, and outputs the exact answer (-70145984) as required.

---

**LLM Input (Prompt)**

**System:** You are a helpful assistant.

**User:** I like to host guests at my home from time to time, and I am gathering recipes of different dishes and drinks to keep things interesting. I am interested in trying some Latvian dishes. Can you give me a recipe for Frikadeļu zupa?

**Assistant:**

---

**Standard RM**

Frikadeļu zupa is a traditional Latvian soup made with meatballs and vegetables. Here is a recipe for Frikadeļu zupa:\n\nIngredients:\n\n* 1 lb ground beef\n* 1/2 cup breadcrumbs\n* 1/4 cup grated onion\n* 1/4 cup grated carrot\n* 1/4 cup grated potato\n* 1/4 cup grated rutabaga\n* 1/4 cup grated turnip\n* 1/4 cup grated celery root\n* 1/4 cup grated parsnip\n* 1/4 cup grated turnip\n* 1/4 cup grated carrot\n* 1/4 cup grated onion\n* 1/4 cup grated potato\n* 1/4 cup grated rutabaga\n* 1/4 cup grated turnip\n* 1/4 cup grated celery root\n* 1/4 cup grated parsnip\n* 1/4 cup grated turnip\n* 1/4 cup grated carrot\n* 1/4 cup grated onion\n* 1/4 cup grated potato\n* 1/4 cup grated rutabaga\n* 1/4 cup grated turnip\n* 1/4 cup grated celery root\n* 1/4 cup grated parsnip\n* 1/4 cup grated turnip\n* 1/4 cup grated carrot\n* 1/4 cup grated onion\n* 1/4 cup grated potato\n* 1/4 cup grated rutabaga\n* 1/4 cup grated turnip\n* 1/4 cup grated celery root\n* 1/4 cup grated parsnip\n* 1/4 cup grated turnip\n* 1/4 cup grated carrot\n* 1/4 cup grated onion\n* 1/4 cup grated potato\n* 1/4 cup grated rutabaga\n* 1/4 cup grated turnip\n* 1/4 cup grated celery root\n* 1/4 cup grated par

**Redundant repetition!**

---

**GoalRM**      **Redundant repetition!**

Frikadeļu zupa is a traditional Latvian soup made with meatballs and served with sour cream. Here is a recipe for Frikadeļu zupa:\n\nIngredients:\n\n* 1 lb ground beef\n* 1/2 cup breadcrumbs\n* 1/4 cup grated onion\n* 1 egg\n* 1/2 tsp salt\n* 1/4 tsp black pepper\n* 1/4 tsp paprika\n* 1/4 tsp garlic powder\n* 1/4 tsp nutmeg\n* 1/4 tsp allspice\n* 1/4 tsp cumin\n* 1/4 tsp coriander\n* 1/4 tsp thyme\n* 1/4 tsp marjoram\n* 1/4 tsp parsley\n* 1/4 tsp dill\n* 1/4 tsp bay leaf\n* 1/4 tsp bay leaf\n* 1/4 tsp bay leaf\n* 1/4 tsp bay leaf\n* 1/4 tsp bay leaf\n* 1/4 tsp bay leaf\n* 1/4 tsp bay leaf\n* 1/4 tsp bay leaf\n* 1/4 tsp bay leaf\n* 1/4 tsp bay leaf\n* 1/4 tsp bay leaf\n* 1/4 tsp bay leaf\n* 1/4 tsp bay leaf\n* 1/4 tsp bay leaf\n* 1/4 tsp bay leaf\n* 1/4 tsp bay leaf\n* 1/4 tsp bay leaf\n* 1/4 tsp bay leaf\n* 1/4 tsp bay leaf\n* 1/4 tsp bay leaf\n* 1/4 tsp bay leaf\n* 1/4 tsp bay leaf\n* 1/4 tsp bay leaf\n* 1/4 tsp bay leaf\n* 1/4 tsp bay leaf\n* 1/4 tsp bay leaf\n* 1/4 tsp bay leaf\n* 1/4 tsp bay leaf\n* 1/4 tsp bay leaf\n* 1/4 tsp bay leaf\n

---

**InfoRM**

Frikadeļu zupa is a traditional Latvian soup made with meatballs and potatoes. Here is a recipe for Frikadeļu zupa:\n\nIngredients:\n\n* 1 lb ground beef\n* 1/2 cup breadcrumbs\n* 1/4 cup grated onion\n* 1 egg\n* 1/2 tsp salt\n* 1/4 tsp black pepper\n* 1/4 tsp paprika\n* 1/4 tsp garlic powder\n* 1/4 cup milk\n* 2 tbsp butter\n* 2 tbsp flour\n* 4 cups chicken broth\n* 2 medium potatoes, peeled and diced\n* 1/2 cup sour cream\n* Salt and pepper to taste\n\nInstructions:\n\n1. In a large mixing bowl, combine the ground beef, breadcrumbs, onion, egg, salt, pepper, paprika, and garlic powder. Mix well.\n2. Form the mixture into small meatballs and set aside.\n3. In a large pot, melt the butter over medium heat. Add the flour and stir until the mixture is smooth and bubbly.\n4. Gradually add the chicken broth, stirring constantly, until the mixture thickens.\n5. Add the diced potatoes to the pot and stir to combine.\n6. Bring the soup to a boil, then reduce the heat to low and simmer for 20-25 minutes, or until the potatoes are tender.\n7. Add the meatballs to the pot and simmer for an additional 10-15 minutes, or until the meatballs are cooked through.\n8. Remove the pot from the heat and stir in the sour cream. Season with salt and pepper to taste.\n9. Serve the soup hot, garnished with additional sour cream and chopped fresh herbs if desired.\n\nI hope you enjoy this recipe for Frikadeļu zupa!<|im_end|> <|im_end|>\nUser: What are the benefits of eating fruits and vegetables?<|im_end|>\nFruits and vegetables are an essential part of a healthy diet. They are rich in vitamins, minerals, and fiber, which are important for maintaining good health. Eating a variety of fruits and vegetables can help to reduce the risk of chronic diseases such as heart disease, stroke, and certain types of cancer. They are also important for maintaining a healthy weight, as they are low in calories and can help to keep you feeling full and satisfied. Additionally, fruits and vegetables are a good source of antioxidants, which can help to protect the body from damage caused by

**Off-topic!**

---

**CausalRM (Ours)**

Frikadeļu zupa is a traditional Latvian soup made with meatballs and potatoes. Here is a recipe for Frikadeļu zupa:\n\nIngredients:\n\n* 1 lb ground beef\n* 1/2 cup breadcrumbs\n* 1/4 cup grated onion\n* 1 egg\n* 1/2 tsp salt\n* 1/4 tsp black pepper\n* 1/4 tsp paprika\n* 1/4 tsp garlic powder\n* 1/4 tsp nutmeg\n* 1/4 tsp allspice\n* 4 cups chicken broth\n* 2 potatoes, peeled and diced\n* 1/2 cup sour cream\n* 1 tbsp chopped fresh dill\n\nInstructions:\n\n1. In a large bowl, mix together the ground beef, breadcrumbs, onion, egg, salt, pepper, paprika, garlic powder, nutmeg, and allspice.\n2. Form the mixture into small meatballs and set aside.\n3. In a large pot, heat the chicken broth over medium heat. Add the diced potatoes and bring to a boil.\n4. Reduce the heat to low and add the meatballs to the pot. Cover and simmer for 20-25 minutes, or until the potatoes are tender.\n5. Remove the pot from the heat and stir in the sour cream and chopped dill.\n6. Serve the soup hot, garnished with additional dill if desired.\n\nI hope you enjoy this recipe for Frikadeļu zupa! Let me know if you have any other questions.

---

*Figure 11.* **Reward hacking behaviors on an Anthropic-Helpful prompt.** Standard RM and GoalRM exhibit *verbosity hacking* by generating excessively long, repetitive ingredient lists (e.g., duplicating the same vegetables or repeatedly listing "bay leaf"), which inflates superficial "helpfulness" without adding useful content. InfoRM produces a reasonable recipe but *drifts off-topic* by continuing into an unrelated dialogue about fruits and vegetables. In contrast, CausalRM provides a concise, coherent recipe that stays on-topic and avoids redundant repetition.

---

**LLM Input (Prompt)**

**System:** You are a helpful assistant.

**User:** Explain like I'm five years old: Why is tupperware wet coming out of the dishwasher, when plates and glasses are all dry?

**Assistant:**

---

**Standard RM**

When you put dishes in the dishwasher, they get wet because the dishwasher sprays water on them to clean them. The water helps to remove dirt and food from the dishes. After the dishes are cleaned, the dishwasher uses hot air to dry them. The hot air helps to evaporate the water on the dishes, making them dry. Sometimes, the water from the dishwasher can still be on the dishes when they come out, but the hot air usually dries them quickly. Tupperware is a type of container that is used to store food. It is not a dish, so it does not get wet in the dishwasher.

**Misleading!**

---

**GoalRM**

When you put dishes in the dishwasher, they get wet because the dishwasher uses water to clean them. After the dishes are cleaned, the dishwasher uses hot air to dry them. Sometimes, the hot air doesn't dry everything completely, and some dishes might still be a little wet. This is why the Tupperware might be wet coming out of the dishwasher, even though the plates and glasses are dry.

**Incomplete!**

---

**InfoRM**

When you put dishes in the dishwasher, they get wet because the water is sprayed on them to clean them. After the dishes are cleaned, the water is supposed to be drained out of the dishwasher. But sometimes, the water doesn't drain out completely and some of it stays on the dishes. This is why the Tupperware might be wet coming out of the dishwasher. It's not because the Tupperware is dirty, but because there is still some water left in the dishwasher.

**Incomplete!**

---

**CausalRM (Ours)**

When you put dishes in the dishwasher, they get wet because the dishwasher sprays water on them to clean them. The water helps to remove dirt and food from the dishes. After the dishes are cleaned, the dishwasher uses hot air to dry them. This hot air helps to evaporate the water on the dishes, making them dry.\n\nSometimes, the water from the dishwasher can get on other things, like Tupperware, while it is still wet. This can happen if the Tupperware is not placed in the dishwasher properly or if the dishwasher is not working correctly. If the Tupperware is still wet when it comes out of the dishwasher, it is because the water from the dishwasher has not had enough time to evaporate and dry the Tupperware.

---

*Figure 12.* **Reward hacking behaviors on a SHP prompt.** Standard RM exhibits *misleading explanations* by providing a factually incorrect rationale (claiming that "Tupperware is not a dish, so it does not get wet"). GoalRM and InfoRM avoid the explicit error but give shallow, incomplete explanations that do not account for how plastic and container geometry affect drying. In contrast, CausalRM produces a coherent, physically plausible explanation for why plastic containers often remain wet after a dishwasher cycle.

