# OpenReview forum: "Factored Causal Representation Learning for Robust Reward Modeling in RLHF"
_ICML.cc/2026/Conference — ICML 2026 regular_

### Official Review · Reviewer_FFwM · 2026-02-18

**Soundness:** 2
**Presentation:** 3
**Significance:** 2
**Originality:** 2
**Overall Recommendation:** 3
**Confidence:** 3

**Summary:**

This paper tackles reward hacking in RLHF, where reward models overfit to spurious correlates such as response length or sycophantic phrasing rather than the underlying qualities humans care about (Introduction; Fig. 1). The proposed method, CausalRM, inserts a lightweight latent bottleneck on top of an LLM backbone and factorizes the backbone embedding into two latents: a “causal” part z_c used by the reward head and a “non-causal” part z_nc intended to capture reward-irrelevant variation (Sec. 3.2; Fig. 2). Training combines a standard Bradley–Terry preference loss with KL-based information bottleneck regularization, a reconstruction loss to prevent degenerate factorizations, and an adversarial head with gradient reversal to discourage z_nc from encoding reward-predictive signals (Sec. 3.3; Eq. 10–15; Alg. 1; Appendix A/B). Experiments on mathematical reasoning (OpenMathInstruct-1 preference data with ID/OOD benchmarks) and open-ended dialogue (Anthropic-HH with several OOD sets) report improved reward-model pairwise accuracy (Table 1), improved downstream PPO-RLHF performance (Table 2–3), and reduced sensitivity to length and a constructed sycophantic-prefix artifact (Fig. 5; Table 4). The paper also includes ablations showing the benefit of factorization, adversarial training, reconstruction, and KL regularization (Table 9–10; Fig. 6).

**Compliance With Llm Reviewing Policy:**

Affirmed.

**Final Justification:**

My final recommendation remains 3. After considering both the paper and the authors’ rebuttal, I think the work is technically clearer and more solid than it initially appeared. In particular, the rebuttal largely addressed my main earlier concerns: it clarified the intended positioning of the method as an invariance-motivated decomposition rather than a strong causal identification claim, provided missing details on OOD preference-label construction, added multi-seed statistics and bootstrap confidence intervals, clarified the backbone embedding/tuning setup, and gave additional evidence on judge robustness and baseline comparisons. These responses improved my confidence in the soundness and clarity of the empirical evaluation, and I appreciate the authors’ careful and constructive engagement.

That said, my final score does not change because my remaining reservation is primarily about contribution strength, originality, and overall significance relative to the bar. The core method is, in my view, a well-motivated but incremental combination of known ingredients—latent factorization, information-bottleneck-style regularization, adversarial information removal via GRL, and reconstruction—applied to reward modeling. The empirical results are generally positive, but the overall gains are moderate, and I do not see a sufficiently strong conceptual, theoretical, or empirical leap to justify acceptance at this venue. So, while I now view the paper as more technically sound and better specified than before, the rebuttal mainly reinforced that the main limitation is not unresolved technical flaws, but rather that the contribution is not strong enough in novelty/significance for acceptance. I hope the authors continue this line of work, as the problem is important and the paper has several useful ideas and analyses that could form the basis of a stronger future submission.

**Key Questions For Authors:**

1) How exactly are OOD preference pairs and labels built for the reward-model accuracy numbers in Table 1 (especially dialogue OOD: MT-Bench and TruthfulQA)? Please specify: (a) how candidate responses are generated (models, decoding parameters), (b) how chosen vs rejected is defined, and (c) whether any judge model is used. If this pipeline is clearly defined, audited, and released, it would substantially increase my confidence in the OOD generalization claims; if not, I would further discount the OOD results.
2) What is the statistical stability of Tables 1–3 across random seeds? Please report mean±std (or confidence intervals) over at least 3 seeds for RM training and RLHF, and provide confidence intervals for win rates in Table 3. If gains remain consistent across seeds, my evaluation would improve; if variance overlaps baselines, the claimed improvements may be noise-level.
3) What is the precise definition of the backbone embedding h used for the bottleneck (layer/token/pooling) and what is the backbone tuning strategy during RM training (fully fine-tuned vs LoRA vs frozen)? This affects both reproducibility and the interpretation of length-sensitivity results (Fig. 5). A clear specification (and ideally code/config release) would strengthen the paper.
4) Baseline fairness: what hyperparameter search was performed for GoalRM and InfoRM, and are implementations aligned with the cited papers? InfoRM’s collapse in math RLHF (Table 2) is surprising. If the authors can demonstrate matched training recipes and/or provide tuned baselines, it would reduce concerns that the reported gains are partly due to under-tuned baselines.

**Limitations:**

Not fully. The paper does acknowledge that it does not provide identifiability guarantees and is mainly guided by an invariance principle (Appendix E), which is good. However, it should more explicitly discuss limitations of (1) LLM-as-judge evaluation for dialogue (Sec. 4.1; Table 3–4; Appendix C), including judge bias and sensitivity to verbosity/politeness, and (2) the dependence of conclusions on how OOD preference labels are constructed (Table 1). I suggest adding a dedicated paragraph clarifying these evaluation limitations and (ideally) adding multi-judge or small human validation experiments to bound the risk.

**Strengths And Weaknesses:**

Strengths:
1) The method provides a clear architectural constraint: the reward head only reads z_c, and an adversarial head with gradient reversal pressures z_nc to be non-predictive of preference labels (Sec. 3.2–3.3; Fig. 2; Eq. 10–15).
2) The paper includes targeted analyses aligned with the stated goal of mitigating spurious correlations, including length sensitivity on math (Fig. 5) and robustness to a sycophantic-prefix artifact on dialogue (Table 4).
3) Ablations are reasonably comprehensive and support the claim that multiple components contribute (factorization, GRL/adversary, reconstruction, and KL terms; Table 9–10; Fig. 6).
4) Implementation details such as module dimensions, key hyperparameters, OpenRLHF version, and approximate training cost are provided (Appendix B; Table 5–8).

Weaknesses:
1) The central “causal” claim is not technically established; the method is better described as an invariance-motivated factorization plus adversarial information removal, without identifiability guarantees (Sec. 3.1–3.3; Appendix E). This weakens the conceptual solidity and risks over-interpretation of results as “causal.”
2) A critical missing detail is how OOD pairwise labels for reward-model evaluation are constructed for datasets that do not naturally provide human pairwise preferences (e.g., MT-Bench, TruthfulQA in Table 1). Missing: candidate generation process, chosen/rejected definition, and whether an LLM judge is used. Impact: it is difficult to audit whether the OOD accuracy improvements truly reflect generalization rather than artifacts of the label construction.
3) The dialogue downstream evaluation relies on a single LLM judge (Qwen3-Max) with one prompt (Sec. 4.1; Appendix C; Table 3–4). Missing: multi-judge checks or any human validation, plus judge parameter details. Impact: win-rate conclusions may reflect judge-specific biases, possibly correlated with length/politeness/verbosity, which interacts with the paper’s thesis.
4) No statistical reliability is reported (no multiple seeds, no variance or confidence intervals), yet many reported gains are modest (often 1–3 points in Table 1–2). Missing: mean±std over seeds or confidence intervals for win rates. Impact: unclear if improvements are robust beyond noise and implementation randomness.
5) Baseline coverage is limited to Standard RM, GoalRM, and InfoRM (Sec. 4.1; Table 1–3), while the introduction cites other relevant anti-hacking methods (e.g., ODIN, length regularization, MMD-based regularization). Missing: comparisons to at least one representative “explicit spurious-control” baseline. Impact: claims of state-of-the-art robustness are not well supported.
6) Reproducibility-critical implementation details remain unclear: what exact backbone embedding h is used (layer/token/pooling), and whether the backbone is fully fine-tuned or partially tuned/frozen during RM training (Sec. 2; Sec. 3.2; Alg. 1). Impact: key results (especially length sensitivity) can depend strongly on these choices.
7) Some baseline behavior is suspicious: InfoRM collapses severely in math RLHF (Table 2) despite being positioned as an anti-hacking method. Missing: evidence of careful baseline hyperparameter tuning or alignment with original implementations. Impact: raises concerns about fairness and whether the reported gains are partly due to weaker baseline configurations.

---

> ### Author Rebuttal · Authors · 2026-03-30
>
> Thank you for the detailed review. Below we address each concern.
>
> **On the causal claim.**
> Please see our response to **Reviewer KM4J**, Sec. **On the causality claim and identifiability**.
>
> **OOD Preference Label Construction.**
> All evaluation datasets are *publicly available*, and we perform *no* additional label construction. We have added the corresponding links in Appendix B.
>
> *(a) Preference pairs:* Math candidates come from GoalRM (CodeLlama-7B, greedy decoding). Dialogue candidates come from RewardBench [1]. For TruthfulQA, we use the public HuggingFace version formatted in preference style, where each question is paired with correct and incorrect reference answers.
>
> *(b) Labels:* Math: chosen=correct solution, rejected=incorrect solution. Dialogue: labels follow original dataset annotations.
>
> *(c) Judge:* No LLM judge was used for OOD evaluation. Labels rely on ground-truth verification or original annotations, avoiding judge bias.
>
> **LLM Judge Details & Multi-Judge Validation.**
>
> *(a) Judge parameters:*
> We use temperature = 0.0, top\_p = 1.0, and max\_new\_tokens = 512. The evaluation prompt is given in Appendix C.
>
> *(b) Multi-judge validation:*
> We additionally evaluate with DeepSeek-V3.2 and Kimi-k2.5. Please see Tables 3 and 4 of [https://anonymous.4open.science/r/8811/CausalRM.pdf](https://anonymous.4open.science/r/8811/CausalRM.pdf), where on OOD evaluation CausalRM achieves average win rates of 58.3/51.7/35.7/44.9 against SFT, Standard RM, GoalRM, and InfoRM under DeepSeek, and 56.7/53.5/34.8/36.3 under Kimi. The overall conclusions remain consistent across judges, suggesting that the observed gains are not specific to Qwen3-Max.
>
> **Statistical reliability across seeds.**
> Tables 1-3 currently report the mean over 3 runs. Following prior work such as GoalRM and InfoRM, we did not include the variance in the main paper. The full Tables 1-3 with mean$\pm$std are provided in Tables 5-7 of [https://anonymous.4open.science/r/8811/CausalRM.pdf](https://anonymous.4open.science/r/8811/CausalRM.pdf).
>
> To further assess statistical reliability for the dialogue results in Table 3, we also conduct paired bootstrap tests for CausalRM against each baseline.
>
> *Setup.* For each baseline, we use paired LLM-judge outcomes on the same test instances across the six dialogue benchmarks and repeatedly resample instances with replacement. For each resample, we compute the preference margin $\Delta$ = (#{Win} - #{Lose}) / N from the perspective of CausalRM, where ties contribute 0. We run 10,000 bootstrap resamples and report the 95\% percentile confidence interval (CI) of $\Delta$. If the CI lies strictly above 0, CausalRM is significantly preferred over the corresponding baseline.
>
> *Results.* Table 8 of [https://anonymous.4open.science/r/8811/CausalRM.pdf](https://anonymous.4open.science/r/8811/CausalRM.pdf) reports the average win rate of CausalRM over the six dialogue datasets together with the 95\% bootstrap CI of the preference margin $\Delta$. CausalRM is consistently preferred over all baselines, achieving average win rates of 61.6\% (vs. SFT), 42.9\% (vs. Standard RM), 33.2\% (vs. GoalRM), and 34.1\% (vs. InfoRM), with corresponding 95\% CIs of $\Delta$ equal to [50.7, 59.7], [17.2, 27.7], [4.2, 14.6], and [6.0, 15.9]. Since all CIs lie strictly above 0, the gains in Table 3 are statistically reliable.
>
> **Baseline coverage.**
> Please see our reply to **Reviewer 3m3j** (Sec. **Expanded baselines**).
>
> **Backbone embedding and tuning details.**
> h is the final-layer hidden state of the last non-padding token from the SFT-initialized RM backbone, i.e., a contextual embedding without pooling. As specified in Eq.(3), this is a standard reward modeling choice. The backbone is fully fine-tuned with all CausalRM heads. We have clarified this in the appendix, and will release the code upon publication.
>
> **Baseline fairness.**
> We were also surprised by the InfoRM collapse on math RLHF. Importantly, our baseline implementations strictly follow the original papers. For InfoRM, we use the Appendix E.2 recommendations from the original paper, including IB dimension 128 and KL coefficient 0.1. For GoalRM, we follow the default hyperparameters reported in its Table 4.
>
> To better understand this discrepancy, we added a diagnostic study of representation behavior during RL training (Fig. 3). Our observations suggest a potential math-specific formatting failure mode: as training progresses, InfoRM tends to generate increasingly short or incomplete solutions, and often omits the required `\boxed{}` final-answer format. Many outputs therefore no longer contain a parsable final answer, which plausibly explains the rapid accuracy drop. This is supported by Fig. 3 and Fig. 5, and we include qualitative examples in Fig. 7.
>
> We hope these responses address your concerns. Thank you again for your thoughtful review.
>
> [1] Lambert et al. "Rewardbench: Evaluating reward models for language modeling." NAACL 2025.

---

> > ### Author Rebuttal · Reviewer_FFwM · 2026-04-02
> >
> > Thank you for the detailed rebuttal. I have read it carefully, and several of my original concerns are now substantially clarified. In particular, the rebuttal helps on (i) the construction of OOD preference pairs/labels for Table 1, (ii) the exact backbone embedding and full fine-tuning setup, (iii) the use of 3 runs together with additional variance / bootstrap analyses, and (iv) the concern that the dialogue conclusions might be specific to a single judge model, since you now report additional judge results.
> >
> > That said, my central concern is only partially resolved. The main remaining issue is the paper’s “causal” framing. The additional discussion is helpful, but the new identifiability argument appears to rely on explicit environment-variation and invariance assumptions that are still not clearly instantiated in the actual preference-only training setup used in the paper. Likewise, the added intervention-style evidence on math is useful, but it seems task-specific and does not yet fully justify a broader causal interpretation across settings. As a result, I still view the method as more convincingly an invariance-motivated factorization / robustness approach than a strongly established causal identification result.
> >
> > I also think the baseline situation is improved but not fully closed. The added ODIN/MMD comparisons are helpful, but they are reported in targeted artifact-specific settings rather than under the same main evaluation protocol as Tables 1–3, so it is still somewhat unclear how comprehensive the empirical comparison is in the paper’s primary setting.
> >
> > My follow-up questions are:
> > 1) Could you clarify more explicitly whether the intended claim of the paper is causal identification, or rather an invariance-motivated decomposition that is useful for reward modeling? This distinction would materially affect my assessment of the paper’s technical solidity and positioning.
> > 2) For the newly added multi-seed results, could you highlight whether the main gains over Standard RM / GoalRM in Tables 1–3 are consistently outside the reported standard deviations (especially for the math RLHF results)? This would help me judge how much weight to place on the quantitative improvements.
> > 3) For the added ODIN/MMD comparisons, could you clarify in the paper exactly what additional supervision they require relative to CausalRM, and whether any more direct subset comparison under the main protocol is possible?
> >
> > Overall, the rebuttal improves my confidence in several parts of the empirical section, but it does not fully remove my concern that the paper’s central causal interpretation is stronger than what is currently established.

---

> > > ### Author Response · Authors · 2026-04-03
> > >
> > > Thank you for the careful follow-up. We are encouraged that several empirical concerns were clarified. Below, we respond to the follow-up questions.
> > >
> > > **On the intended claim.**
> > > We appreciate the distinction, and we would like to make our intended positioning more explicit from the following perspectives:
> > >
> > > - **Core claim: CausalRM is an invariance-motivated decomposition for robust reward modeling.**
> > >   We agree that our approach should be positioned as a method centered on the causal invariance principle (Fig. 1 and Eq. 5), which is useful for mitigating reward hacking, rather than as a theoretical guarantee of full identification in RLHF. This is also what we intended to convey in Appendix E.
> > >
> > > - **Our previous discussion of identifiability was primarily meant to justify why the use of the term “causal” is appropriate here.**
> > >   We summarize the reasons again here:
> > >   - this use of causal invariance is consistent with prior works;
> > >   - the identifiability discussion was intended as conceptual grounding, i.e., to explain why separating reward-relevant and reward-irrelevant variation is a meaningful modeling direction under a causal-invariance view;
> > >   - interventions on both factors provide supportive evidence that the decomposition is causally meaningful in practice; and
> > >   - the empirical results further show that decomposing representations leads to improved robustness and downstream performance.
> > >
> > > - **Accordingly, we view the contribution of the paper as a practical step toward causally robust reward modeling, supported primarily by empirical evidence.**
> > >   We hope this can also motivate future work that develops stronger causal identifiability guarantees for reward modeling.
> > >
> > > **On the multi-seed stability.**
> > > Yes, the main gains over Standard RM and GoalRM are consistently outside the reported standard deviations, indicating robust improvements rather than noise:
> > >
> > > - **Math (Tables 1 & 2).** The performance gains exceed the variance across both reward modeling and RLHF. For instance, in Table 2 (Math RLHF ID Avg), CausalRM achieves $74.0 \pm 0.5$, while GoalRM achieves $72.5 \pm 0.6$. The lower bound of CausalRM ($73.5$) remains above the upper bound of GoalRM ($73.1$), ensuring non-overlapping error bars. Similar trends are observed in OOD settings ($89.6 \pm 0.4$ vs. $88.5 \pm 0.5$) and Reward Modeling accuracy (Table 1), confirming the stability of our method.
> > > - **Dialogue (Table 3).** In addition to the mean$\pm$std reported in Table 3, our paired bootstrap tests show 95\% CI for the preference margin strictly above 0 against all baselines (e.g., [17.2, 27.7] vs. Standard RM). This further validates that the win-rate improvements are statistically significant.
> > > - **Conclusion.** Across all tasks, the standard deviations are small relative to the performance gaps (e.g., gain of 1.5% vs. std of 0.5% in Math RLHF). This consistency suggests that the improvements are driven by the methodological advantages of CausalRM rather than random seed variance.
> > >
> > > **On the ODIN/MMD comparisons and additional supervision.**
> > > We clarify this in three points:
> > >
> > > - **Additional supervision required by ODIN/MMD.** Taking MMD as an example, it requires explicitly specified artifact attributes (e.g., response-length bins, sycophantic prefixes, concept tags, or demographic groups) in order to compute the distributional-distance constraint across its four bias scenarios. ODIN similarly requires a decomposed reward supervision signal (e.g., separate quality and length components) to train its disentangled heads. For detailed settings, we kindly recommend referring to the original MMD paper.
> > >
> > > - **Why targeted artifact-specific settings are necessary.** These methods assume the spurious variables are *known a priori*. However, in real-world preference datasets, such attributes are typically unknown or ambiguous. For example, in math tasks, the dataset does not contain explicit artifact labels, making a direct comparison infeasible without construction. In dialogue tasks, attributes like length may also correlate with quality in some contexts. Therefore, constructing known spurious variables is necessary for a fair evaluation of artifact-specific methods.
> > >
> > > - **Direct comparison under main protocol.** We further provide a direct comparison on math tasks where length is treated as a known spurious factor. We show the results below:
> > >
> > > | **Method** | **ID Avg.** | **OOD Avg.** |
> > > |---|---:|---:|
> > > | ODIN (length-specific) | 68.8 | 83.2 |
> > > | MMD (length-specific) | 69.3 | 83.6 |
> > > | **CausalRM (Ours)** | **70.1** | **85.6** |
> > >
> > > **Table:** Reward model accuracy (\%) on math tasks with length bias mitigation.
> > >
> > > This demonstrates the advantage of our general factorization approach over artifact-specific regularization.
> > >
> > > We sincerely appreciate your constructive feedback throughout this process, which has helped us improve the clarity and rigor of our paper. We will incorporate all the above discussions into the final revision.

---

### Official Review · Reviewer_3m3j · 2026-03-12

**Soundness:** 2
**Presentation:** 3
**Significance:** 3
**Originality:** 3
**Overall Recommendation:** 3
**Confidence:** 3

**Summary:**

This work proposed the method to decompose the model's contextual embedding into causal and non-causal factors, in order to mitigate reward hacking problems. The authors introduced a structured factorization framework with theoretical analysis. Experiments showed improved robustness and generalization.

**Compliance With Llm Reviewing Policy:**

Affirmed.

**Final Justification:**

Please see my reply to rebuttal.

**Key Questions For Authors:**

How realistic are the identifiability assumptions in practical settings?
Can you provide deeper ablations to isolate the impact of each modeling component in the factorization framework?

**Limitations:**

Yes

**Strengths And Weaknesses:**

Strengths
1. The motivation is clear to combine representation learning and causal inference to improve reward model quality.
2.  The factorization framework is based on structural causal modeling.
3. Theoretical analysis is helpful to clarify when the recovery of causal factors is possible.

Weakness:
1. The identifiability guarantees has strong assumptions. This makes it less practical in real-life situation.
2. The set of baselines can be expanded more.
3. Should conduct more ablation study on studying the impact of each modeling component.

---

> ### Author Rebuttal · Authors · 2026-03-30
>
> Thank you for the careful review and for recognizing the motivation, factorization framework, and theoretical analysis. Below we address your concerns.
>
> **Identifiability assumptions and practicality.**
> Our CausalRM is motivated by one assumption that reward-relevant factors can be separated from spurious factors. This assumption is common in Causal RL, which has already seen successful applications in many real-world domains including healthcare and robotics [1,2]. In the LLM setting, recent causal RL work such as DEPTH [3] further suggests that factorization-based designs can be practically useful in enterprise-scale relation extraction. This indicates the potential real-world applicability of CausalRM. Our results in Tables 1--3 and Sec. 4.3 further support this point: CausalRM achieves strong reward modeling performance, robust downstream RLHF behavior, and improved OOD generalization on real-world mathematical and dialogue tasks. Regarding identifiability, we show that under explicit assumptions, the reward-relevant subspace is identifiable up to invertible transformation. For the full discussion, please see our response to **Reviewer KM4J**, Sec. **On the causality claim and identifiability.**
>
> **Expanded baselines.**
> Thank you for the suggestion. To address this concern, we have added direct comparisons with ODIN and MMD-based regularization under the targeted bias settings used in Sec. 4.3, namely the length-bias and sycophantic-bias evaluations. On length bias, CausalRM achieves the lowest sensitivity with $\sigma_{\text{len}}=0.03$, compared to 0.12 for ODIN and 0.09 for MMD. On sycophantic artifacts, CausalRM also performs best, with average pairwise accuracy of 70.6/61.2 on ID/OOD, compared to 66.7/57.5 for ODIN and 67.1/57.9 for MMD. Full results are provided in Fig. 2 and Table 2 of [https://anonymous.4open.science/r/8811/CausalRM.pdf](https://anonymous.4open.science/r/8811/CausalRM.pdf). These results show that while artifact-specific methods are competitive when the target bias is known in advance, CausalRM remains stronger overall and has the additional advantage of not requiring task-specific artifact supervision.
>
> The main reason these methods were not included in Tables 1-3 is that they assume a different supervision setting from our primary experiments. In particular, methods such as ODIN or MMD-based regularization require the target spurious variable to be explicitly specified (and often annotated or controllably perturbed) during training. For example, to study sycophantic bias with such methods, one must first construct a dataset with a known artifact such as a prefix perturbation, as we did in Sec. 4.3. By contrast, our main experiments are conducted on standard RLHF datasets, where the exploited spurious variables are not explicitly labeled and cannot be exhaustively enumerated in advance. This is also why we position CausalRM as a more general latent-factor approach for mitigating unknown spurious correlations, rather than a method tailored to a pre-specified artifact.
>
> **Ablation on modeling components.**
> Our factorization framework has three main components: the causal reward head, the reconstruction head, and the adversarial (GRL) head. We have already included component-wise ablations in Appendix D, where we separately study the effects of removing each component (corresponding to the ablation variants summarized around Eq. 10). We also investigate additional variants removing the KL losses. In addition, Standard RM can be viewed as the case with only the preference loss. These results consistently support the contribution of each component.
>
> To address your concern, we have also added sensitivity analyses on key hyperparameters, including the GRL coefficient and KL coefficients. Please see Fig. 1 of [https://anonymous.4open.science/r/8811/CausalRM.pdf](https://anonymous.4open.science/r/8811/CausalRM.pdf). The results indicate that CausalRM remains stable across a reasonable range of settings, with consistent overall behavior. If you have specific additional ablation configurations in mind, we would be happy to provide the corresponding results.
>
> Thank you again for the constructive feedback. We hope these clarifications address your concerns.
>
> [1] Schölkopf et al. ``Toward causal representation learning.'' (2021).
>
> [2] Zeng et al. ``A survey on causal reinforcement learning.'' (2024).
>
> [3] Yang et al. ``DEPTH: Hallucination-Free Relation Extraction via Dependency-Aware Sentence Simplification and Two-tiered Hierarchical Refinement.'' (2025).

---

> > ### Author Rebuttal · Reviewer_3m3j · 2026-04-03
> >
> > The rebuttal does not address the core issue of strong and unrealistic identifiability assumptions, and still lacks convincing justification in RLHF settings (also acknowledged in the paper). Added experiments are helpful but insufficient to change the original assessment. Hence I will keep my score.

---

> > > ### Author Response · Authors · 2026-04-03
> > >
> > > Thank you for your thoughtful follow-up. We appreciate your continued engagement with our work. However, there may be some ambiguity regarding the intended scope of our causal claims and the role of the identifiability discussion. To make our positioning fully clear, we would like to restate the following points:
> > >
> > > - **Our intended claim is invariance-driven factorization for robust reward modeling, not full identifiability in RLHF.**
> > >   We would like to clarify that establishing full causal identification from observational preference data is a known open challenge, and providing such a theoretical guarantee is **not** the primary focus of this study. As emphasized throughout the paper and rebuttal, CausalRM is positioned as an *causal invariance-motivated decomposition for robust reward modeling* (centered on Fig. 1 and Eq. 5), rather than as a formal identifiability theorem. The identifiability discussion is therefore meant as *theoretical motivation*: it makes explicit the conditions under which the factorization would be theoretically justified, and conceptually grounds why separating reward-relevant and reward-irrelevant variation is a principled direction.
> > >
> > >   Regarding the concern that the paper "still lacks convincing justification in RLHF settings," this may partly arise from different expectations about the strength of the causal claim being made. If the goal were to establish a full RLHF-specific causal-identification theorem, then we agree that substantially stronger assumptions and a different theoretical scope would be needed. For the claim pursued in this paper, however, we view the relevant justification in RLHF as primarily operational and empirical: consistent OOD robustness, reduced sensitivity to spurious attributes, and direct intervention-style analyses on reward-determining factors. These are the main forms of evidence we provide. Under this intended scope, the paper is not claiming to resolve identifiability, but rather to show that an invariance-driven factorization can lead to empirically robust and practically useful reward models within standard RLHF pipelines.
> > >
> > > - **The underlying assumption is not unrealistically strong.**
> > >   CausalRM relies on the foundational assumption that reward-relevant factors can be disentangled from spurious ones. This factorization assumption is standard in causal RL, and has been successfully operationalized in practical domains [1-4]. Importantly, this premise is also increasingly adopted in RLHF: recent approaches such as RMM [5], invariance-based regularization [6,7], backdoor adjustment [8], and template-based factorization [9] also build on the idea that reward-relevant information should be separated from spurious correlations. CausalRM follows this same general premise, but does so through an implicit latent factorization without requiring explicit artifact annotations.
> > >
> > > - **The contribution is practical and empirically grounded.**
> > >   We view CausalRM as a *practical step toward causally robust reward modeling*, supported primarily by empirical evidence. The method is designed, trained, and evaluated within standard RLHF pipelines on real-world mathematical and dialogue datasets. The consistent robustness gains and marked reduction in sensitivity to spurious artifacts directly demonstrate its practical utility in realistic alignment settings.
> > >
> > > We hope this clarification makes our intended scope fully explicit. We thank you again for the thoughtful feedback.
> > >
> > > [1] Schölkopf et al. ``Toward causal representation learning.'' (2021).
> > >
> > > [2] Huang et al. ``Action-sufficient state representation learning for control with structural constraints.'' (2022).
> > >
> > > [3] Liu et al. ``Learning world models with identifiable factorization.'' (2023).
> > >
> > > [4] Zeng et al. ``A survey on causal reinforcement learning.'' (2024).
> > >
> > > [5] Liu et al. ``Rrm: Robust reward model training mitigates reward hacking.'' (2024).
> > >
> > > [6] Ovinnikov et al. ``Learning causally invariant reward functions from diverse demonstrations. '' (2024).
> > >
> > > [7] Wang et al. ``Beyond reward hacking: Causal rewards for large language model alignment.'' (2025).
> > >
> > > [8] Song et al. ``Causal reward adjustment: Mitigating reward hacking in external reasoning via backdoor correction.'' (2025).
> > >
> > > [9] Yang et al. ``DEPTH: Hallucination-Free Relation Extraction via Dependency-Aware Sentence Simplification and Two-tiered Hierarchical Refinement.'' (2025).

---

### Official Review · Reviewer_KM4J · 2026-03-13

**Soundness:** 3
**Presentation:** 2
**Significance:** 3
**Originality:** 2
**Overall Recommendation:** 4
**Confidence:** 3

**Summary:**

This paper addresses the classic reward hacking problem in RLHF by proposing CausalRM, a causal representation learning framework. It employs a VAE architecture to factorize an LLM's contextual representations into causal and non-causal factors. A composite loss function, whose core is the adversarial training on non-causal factors via a Gradient Reversal Layer (GRL), enforces their disentanglement. Evaluated on mathematical reasoning and open-domain dialogue tasks, CausalRM outperforms baselines consistently in both in-distribution (ID) and out-of-distribution (OOD) settings.

**Compliance With Llm Reviewing Policy:**

Affirmed.

**Final Justification:**

This paper propose CausalRM, a causal representation learning framework that mitigates reward hacking in RLHF. It decomposes LLM representations into causal and non-causal components via a VAE with GRL-based adversarial training, preventing exploitation of shortcuts like answer length or sycophancy.

The rebuttal satisfactorily addresses all initial concerns. New intervention experiments provide causal evidence, targeted baselines (ODIN, MMD) and a clear reward-length sensitivity curve (σ=0.03) demonstrate superior debiasing, and hyperparameter analysis alleviates stability worries about GRL.

Based on these substantial results, I have revised my score from 3 (Weak reject) to 4 (Weak accept).

**Key Questions For Authors:**

See the "Weaknesses".

**Limitations:**

yes

**Strengths And Weaknesses:**

**Strengths**

1.  The work insightfully reframes the RLHF reward hacking problem into one of "causal invariance" and "representation disentanglement." The integrated design combining the information bottleneck principle with GRL-based adversarial training is mathematically sound and intuitively coherent.
2.  The empirical evaluation is thorough, demonstrating consistent performance gains. Appendix D provides a high-quality ablation study that solidly validates the framework's design.

**Weaknesses**

1.  Caution Against Overstating "Causality" Without Identifiability Guarantees: Under purely observational data (only preference labels, without interventions, multiple environments, or proxies for confounders), the identifiability of causal factors via non-linear ICA is mathematically unguaranteed. While Appendix E honestly acknowledges this limitation, it is recommended to soften causal claims throughout the manuscript to more accurately reflect the method's reliance on empirical invariance rather than proven identifiability.
2.  Omission of Relevant Baselines: The introduction cites state-of-the-art works specifically targeting length bias (e.g., ODIN) and length regularization methods. Their absence from the main experimental comparisons (Tables 1, 2, 3) is a notable gap, weakening the comprehensiveness of the empirical validation.
3.  Pivotal Ablation Study Should be in Main Text: Given the complexity of the loss function and the number of hyperparameters, the ablation study plays a critical role in validating the design. Its findings are too central to the methodological argument to be relegated to the appendix and should be integrated into the main body.
4.  Need for More Validation on Training Stability: The stability of the GRL-based adversarial training requires more empirical validation. Providing analysis (e.g., convergence curves, sensitivity to hyperparameters like the GRL coefficient) would strengthen confidence in the method's practical robustness.

---

> ### Author Rebuttal · Authors · 2026-03-30
>
> Thank you for the constructive review. Below we address each concern.
>
> **On the causality claim and identifiability.**
> We use *causal* in two senses: (1) a *causal view* of reward hacking (Fig. 1), where spurious attributes create non-causal pathways; and (2) a *causal invariance objective* (Eq. 5) enforcing insensitivity to reward-irrelevant variations. This usage aligns with recent work framing reward hacking via causal invariance [1,2].
>
> *Theoretical justification.* We agree that causal recovery from observational preference data requires explicit assumptions. Nevertheless, our objective (Eq. 10, which aims to preserve reward-relevant information while suppressing dependence on spurious factors) is grounded in the theoretical guarantee that the reward-relevant subspace is identifiable up to an invertible transform under the following setting:
>
> Consider an observation $o=g(z^c,z^{nc})$, where $z^c\in\mathcal Z^c$ are reward-relevant factors and $z^{nc}\in\mathcal Z^{nc}$ are spurious factors. Assume the environments $e$ affect the data distribution only through $z^{nc}$, while the reward depends only on $z^c$, i.e.,
>
> $r \perp z^{nc}\mid z^c,\qquad p_e(r\mid z^c)=p(r\mid z^c),\ \forall e\in\mathcal E.$
>
> Let a learned representation $\tilde z^c=\phi(o)=\bar\phi(z^c,z^{nc})$. We further assume (A1) sufficient environment variability [3]: whenever an event in latent space cannot be written as a product $B\times\mathcal Z^{nc}$ (i.e., it entangles $z^c$ with $z^{nc}$), its probability changes across some environments; and (A2) representation invariance [4]: $\tilde z^c\perp e$.
>
> Then, $\bar\phi$ cannot depend on $z^{nc}$: if it did, then there would exist a preimage set of $\tilde z^c$ that entangles $z^c$ and $z^{nc}$, whose probability would necessarily vary across environments by (A1), contradicting (A2). Hence $\tilde z^c=\psi(z^c)$.
>
> Finally, if $z^c$ is minimal sufficient for predicting $r$ and $\tilde z^c$ is also sufficient, then $\psi$ must be almost everywhere one-to-one. Otherwise $\tilde z^c$ would be a strictly coarser sufficient statistic than $z^c$, contradicting the minimal sufficiency of $z^c$. Thus, the reward-relevant subspace is identifiable up to an invertible transform.
>
> *Intervention-based evidence.*
> We further added a direct intervention analysis to test whether the learned $z^c$ behaves like a causal factor. In math tasks, the final boxed answer is the most direct reward-determining variable (Fig. 7). We therefore construct two controlled interventions: (i) *correct $\to$ wrong*, replacing the boxed answer with an incorrect one, and (ii) *wrong $\to$ correct*, replacing an incorrect boxed answer with the correct one. If a reward model captures this factor, its reward should decrease in (i) and increase in (ii).
>
> Table 1 of [https://anonymous.4open.science/r/8811/CausalRM.pdf](https://anonymous.4open.science/r/8811/CausalRM.pdf) summarizes the results. CausalRM consistently shows the expected directional change under both interventions, with larger and more stable reward deltas (-0.76 for correct$\to$wrong and +0.77 for wrong$\to$correct) than the baselines, together with much higher reward decrease/increase ratios (86.1\%/86.6\%). This suggests that CausalRM is more sensitive to interventions on the true reward-determining factor rather than superficial correlates.
>
> Additionally, Sec. 4.3 already shows that CausalRM is markedly less sensitive to length and sycophantic artifacts than strong baselines. Together, these results support the intended pattern: CausalRM is *sensitive to causal changes* but *invariant to spurious ones*.
>
> Overall, our ``causal'' interpretation is supported by (i) an invariance-based definition, (ii) an identifiability argument under explicit assumptions, (iii) intervention evidence on reward-determining factors, and (iv) consistent gains in practical alignment performance. We have revised Sec. E to clarify these scopes.
>
> **Omission of relevant baselines.**
> Please see our reply to **Reviewer 3m3j** (Sec. **Expanded baselines**).
>
> **Ablation study.**
> Thank you for the suggestion. We have revised the paper to integrate the key ablation findings into the main text.
>
> **Training stability.**
> We have added sensitivity analyses on key hyperparameters, including the GRL coefficients. Please see Fig. 1 of [https://anonymous.4open.science/r/8811/CausalRM.pdf](https://anonymous.4open.science/r/8811/CausalRM.pdf). The consistent overall behavior show that CausalRM remains stable across a reasonable range of settings.
>
> Thank you again for the thoughtful feedback. We hope these clarifications and added results address your concerns.
>
> [1] Liu et al. ``Rrm: Robust reward model training mitigates reward hacking.'' (2024)
>
> [2] Wang et al. ``Beyond reward hacking: Causal rewards for large language model alignment.'' (2025)
>
> [3] Kong et al. ``Partial identifiability for domain adaptation.'' (2023)
>
> [4] Schölkopf et al. ``Toward causal representation learning.'' (2021)

---

> > ### Author Rebuttal · Reviewer_KM4J · 2026-04-03
> >
> > Thank you for your comprehensive and convincing rebuttal. The additional clarifications and data you provided have fully addressed my concerns. I will therefore update my score to reflect this positive change.

---

> > > ### Author Response · Authors · 2026-04-03
> > >
> > > Thank you very much for taking the time to read our rebuttal and for your positive follow-up comment. We are greatly encouraged that the additional clarifications and results have addressed your concerns!
> > >
> > > We also sincerely appreciate your constructive review, which has helped us improve the clarity and presentation of the paper substantially. We will incorporate all of the above discussions into the final version. Thank you again for your time and thoughtful feedback.

---

### Official Review · Reviewer_PkB2 · 2026-03-13

**Soundness:** 3
**Presentation:** 3
**Significance:** 3
**Originality:** 3
**Overall Recommendation:** 4
**Confidence:** 4

**Summary:**

This paper introduces CausalRM, a framework designed to combat reward hacking in RLHF. CausalRM addresses the issue of non-causal factors, such as response length and sycophantic bias, by separating causal and non-causal components in the model's contextual embedding. Extensive experiments show that CausalRM outperforms some state-of-the-art models, exhibiting better robustness and alignment with human preferences, particularly in out-of-distribution scenarios.

**Compliance With Llm Reviewing Policy:**

Affirmed.

**Final Justification:**

The author has addressed my concerns. I will maintain my positive score.

**Key Questions For Authors:**

NA

**Limitations:**

yes

**Strengths And Weaknesses:**

**Strengths:**

By ensuring reward predictions depend only on causal components, CausalRM addresses a key challenge in RLHF, where models often exploit irrelevant features. The addition of an adversarial head to suppress non-causal influences further strengthens the model's robustness. Moreover, experimental results across multiple domains are convincing, showing that CausalRM improves reward model accuracy and enhances RLHF performance, especially when spurious correlations are present.

**Weaknesses:**

The paper could benefit from a deeper exploration of causal invariance limitations, especially in complex tasks like multi-turn dialogues, where non-causal factors may fluctuate more. Some assumptions about the stability of causal and non-causal factors may not hold in all domains, and a more detailed discussion on the framework’s real-world applicability and scalability would be beneficial. Further validation in diverse and nuanced environments is needed to fully assess the framework's robustness.

---

> ### Author Rebuttal · Authors · 2026-03-30
>
> Thank you for your thoughtful review and positive assessment of our paper. We appreciate your constructive feedback on causal invariance and applicability. Below we address your concerns.
>
> **Causal invariance in complex tasks.** CausalRM has the potential to be extended to settings with varying non-causal factors. One possible approach is to treat reward-relevant causal factors as the invariant component and use domain-specific latent variables to capture non-causal fluctuations. Since reward prediction is based only on the causal component, variations in the non-causal part need not affect the learned reward model. Similar ideas have proven effective in domain adaptation [1,2]. We have added a discussion of this future direction in the appendix.
>
> **Assumptions and real-world applicability.**
> Our CausalRM is motivated by one assumption that reward-relevant factors can be separated from spurious factors. This assumption is common in Causal RL, which has already seen successful applications in many real-world domains including healthcare and robotics [3,4]. In the LLM setting, recent causal RL work such as DEPTH [5] further suggests that factorization-based designs can be practically useful in enterprise-scale relation extraction. This indicates the potential real-world applicability of CausalRM. Our results in Tables 1--3 and Sec. 4.3 further support this point: CausalRM achieves strong reward modeling performance, robust downstream RLHF behavior, and improved OOD generalization on real-world mathematical and dialogue tasks.
>
> Regarding scalability, CausalRM only adds lightweight heads and a GRL module without modifying the main backbone, introducing negligible additional overhead. In our experiments, the training time is comparable to standard reward models. For example, on mathematical tasks, standard RM training takes 6.4 hours on average, while CausalRM takes 6.5 hours.
>
> To further verify scalability, we tested across different model sizes on math tasks, with results shown below:
>
> | **Model** | **GSM8K** | **MATH** | **Algebra222** | **GSM-Hard** | **ASDiv** | **MAWPS** | **SVAMP** | **RM Train Time** |
> |---|---:|---:|---:|---:|---:|---:|---:|---:|
> | Qwen2.5-Math-1.5B | 70.0 | 51.8 | 80.1 | 59.5 | 88.2 | 86.2 | 85.4 | 0.8h |
> | Qwen2.5-Math-7B | 81.7 | 58.4 | 89.9 | 66.2 | 89.9 | 92.2 | 89.6 | 6.5h |
> | Qwen2.5-Math-72B | 80.6 | 63.5 | 91.6 | 69.8 | 92.6 | 94.3 | 89.4 | 26.5h |
>
> **Table:** Scalability of CausalRM on Mathematical Reasoning (Pairwise Accuracy \%).
>
> We have added the corresponding discussion in the appendix.
>
> **Validation in diverse environments.** To assess robustness beyond standard settings, we have already included evaluations in several diverse and nuanced environments. In terms of diversity, CausalRM is tested on two distinct real-world domains, mathematical reasoning and open-ended dialogue, with results on both reward modeling and downstream RLHF reported in Tables 1-3. In terms of nuance, we further study challenging spurious factors that commonly induce reward hacking, including response length and sycophantic bias, as analyzed in Sec. 4.3 (Fig. 5 and Table 4). Together, these experiments provide evidence that CausalRM remains effective across different environments. If you have specific scenarios in mind, we would be very happy to further discuss them.
>
> Thanks again for your time and valuable comments. We hope our responses address your concerns. Please let us know if you have further questions.
>
> [1] Huang et al. ``Adarl: What, where, and how to adapt in transfer reinforcement learning.'' (2021).
>
> [2] Yao et al. ``Temporally disentangled representation learning.'' (2022).
>
> [3] Schölkopf et al. ``Toward causal representation learning.'' (2021).
>
> [4] Zeng et al. ``A survey on causal reinforcement learning.'' (2024).
>
> [5] Yang et al. ``DEPTH: Hallucination-Free Relation Extraction via Dependency-Aware Sentence Simplification and Two-tiered Hierarchical Refinement.'' (2025).

---

> > ### Author Rebuttal · Reviewer_PkB2 · 2026-04-04
> >
> > Thank you for your response. I have no additional questions about this submission. I will maintain my positive score.

---

> > > ### Author Response · Authors · 2026-04-04
> > >
> > > Thank you very much for your follow-up comment and for taking the time to read our rebuttal. We are glad that our response has addressed your concerns.
> > >
> > > We also sincerely appreciate your constructive review and positive assessment, which helped us improve the paper. Thank you again for your time and feedback.

---

### Decision · Program_Chairs · 2026-04-30

**Decision:**

Accept (regular)

**Comment:**

Reviewers agreed that the paper addresses an important problem in RLHF and that the empirical results are consistently strong across reward modeling, downstream RLHF, and analyses of spurious correlations such as length and sycophantic bias. In particular, reviewers found the empirical section thorough after rebuttal, with the added baselines, ablations, stability analysis, and clarification of evaluation details strengthening confidence in the results.

The main point of disagreement concerned the strength of the paper's causal claims. In the discussion, however, the authors clarified that the intended scope is an invariance-motivated factorization approach for robust reward modeling rather than a claim of full causal identification in RLHF. With that clarification, I do not view the identifiability issue as a blocking soundness concern, though I agree that the final version should consistently maintain this more careful positioning.

The strongest remaining concern is about originality and significance. Several reviewers noted that the method combines familiar ingredients, and that the overall contribution is somewhat incremental and architecturally heavy. I agree this limits the paper's impact. However, on balance, I find the paper technically solid, empirically well supported, and likely useful to researchers working on robust reward modeling and reward hacking in RLHF. This makes it a borderline but positive paper for the conference.
For the final version, the authors should clearly tone down the causal framing throughout, present the method as primarily invariance-motivated, and incorporate the rebuttal clarifications on evaluation protocol, statistical reporting, and the supervision differences in comparisons to artifact-specific baselines. So, all in all, I would recommend acceptance.